# VinePPO: Unlocking RL Potential For LLM Reasoning Through Refined Credit Assignment

## Abstract

Large language models (LLMs) are increasingly applied to complex reasoning tasks that require executing several complex steps before receiving any reward. Properly assigning credit to these steps is essential for enhancing model performance. Proximal Policy Optimization (PPO), a state-of-the-art reinforcement learning (RL) algorithm used for LLM finetuning, employs value networks to tackle credit assignment. However, value networks face challenges in predicting the expected cumulative rewards accurately in complex reasoning tasks, often leading to high-variance updates and suboptimal performance. In this work, we systematically evaluate the efficacy of value networks and reveal their significant shortcomings in reasoning-heavy LLM tasks, showing that they barely outperform a random baseline when comparing alternative steps. To address this, we propose VinePPO, a straightforward approach that leverages the flexibility of language environments to compute unbiased Monte Carlo-based estimates, bypassing the need for large value networks. Our method consistently outperforms PPO and other RL-free baselines across MATH and GSM8K datasets with fewer gradient updates (up to 9x), less wall-clock time (up to 3.0x). These results emphasize the importance of accurate credit assignment in RL finetuning of LLM and demonstrate VinePPO's potential as a superior alternative.

## 1 Introduction

Large language models (LLMs) are increasingly used for tasks requiring complex reasoning, such as solving mathematical problems (OpenAI, 2024), navigating the web (Zhou et al., 2024), or editing large codebases (Jimenez et al., 2024). In these settings, LLMs often engage in extended reasoning steps, executing multiple actions to arrive at a solution. However, not all steps are equally important—some contribute significantly, while others are irrelevant or detrimental. For example, in Figure 1.a, only step $s_2$ provides a key insight. Indeed, most reasoning steps generated by a model do not affect the chance of it solving the problem (Figure 1.b). Identifying the contribution of each action is crucial for improving model performance. However, this is inherently difficult due to the significant delay between actions and their eventual effect. This issue, known as the *credit assignment problem*, is a core challenge in reinforcement learning (RL, Sutton and Barto 1998).

Proximal Policy Optimization (PPO, Schulman et al. 2017; Ouyang et al. 2022), a state-of-the-art algorithm for RL-based finetuning of LLMs (Xu et al., 2024; Ivison et al., 2024; Chang et al., 2023), tackles credit assignment using a value network (or critic). This network, typically a separate model initialized from a pretrained checkpoint, is trained during PPO finetuning to estimate the expected cumulative rewards (or value) of an intermediate action. In Figure 1.b, an ideal value network would assign high value to step $s_2$ and subsequent steps, where the model predicted a critical action. PPO uses these value estimates to measure the *advantage* of each action and update the model accordingly.

Accurately modeling value—predicting future rewards from an incomplete response—requires the value network to understand both the space of correct solutions (the very task the policy model is trying to learn) and predict the model's future behavior, both of which are inherently challenging. In fact, there are hints in the literature that standard PPO implementations for LLMs have inaccurate

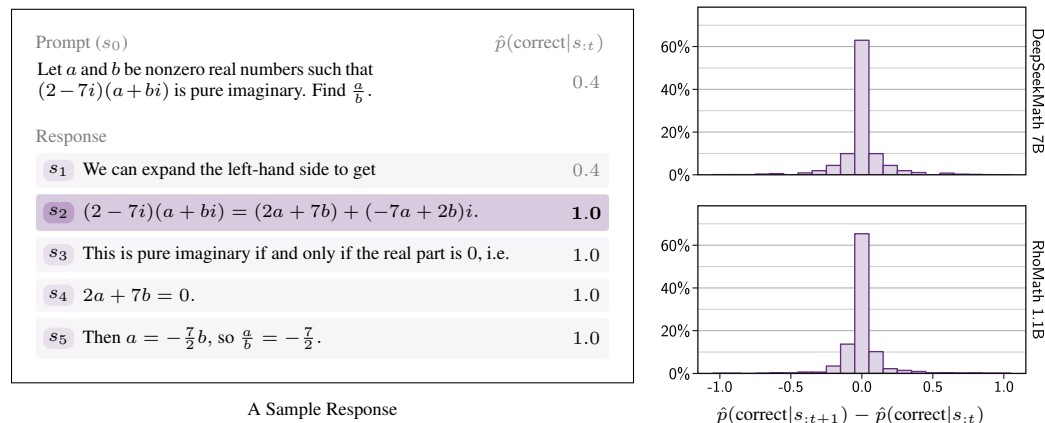

Figure 1: **(Left)** A response generated by the model. The notation $\hat{p}(\text{correct}|s_{:t})$ represents the estimated probability of successfully solving the problem at step $t$. Here, only step $s_2$ is critical; after this, the model completes the solution correctly. **(Right)** The delta in probability of successful completion between response steps. Most steps show little or no advantage over the preceding step.

value estimations. Ahmadian et al. (2024) and Trung et al. (2024) find that value networks often serve best as just a baseline in policy gradient[1]. Shao et al. (2024) show that the value network can be replaced by averaging rewards of responses to a given problem without degradation in performance. Since errors in value estimation can lead to poor credit assignment and negatively impact convergence and performance (Greensmith et al., 2001), a natural question to ask is: *how accurately do value networks actually perform during LLM finetuning?* If we could improve credit assignment, to what extent would it enhance LLM performance? While recent studies (Hwang et al., 2024; Setlur et al., 2024) have begun to highlight the importance of identifying incorrect reasoning steps and incorporating them via ad-hoc mechanisms in "RL-free" methods (Rafailov et al., 2023), the broader question of how improving credit assignment might boost RL fine-tuning for LLMs remains open.

In this work, we evaluate the standard PPO pipeline in mathematical reasoning tasks across various model sizes. We find that value networks consistently provide inaccurate estimates and struggle to rank alternative steps correctly, suggesting that current PPO finetuning approaches for LLMs operate without effective credit assignment. To address this issue and illustrate the effect of accurate credit assignment, we propose VinePPO (Figure 2). Instead of relying on value networks, VinePPO computes *unbiased* value estimates of intermediate states by using independent Monte Carlo (MC) samples and averaging their respective return. This straightforward modification to PPO takes advantage of a special property of the language environment: the ability to easily reset to any intermediate state along the trajectory.

VinePPO consistently outperforms standard PPO and "RL-free" baselines, especially on more challenging datasets. Despite its slower per-iteration speed, it reaches and surpasses PPO's peak performance with *fewer gradient updates*, resulting in *less wall-clock time* and *lower KL divergence* from the base model. Our findings highlight the importance of precise credit assignment in LLM finetuning and establishes VinePPO as a straightforward alternative to value network-based approaches.

Our contributions are as follows:

- We demonstrate the suboptimal credit assignment in standard PPO finetuning by analyzing the value network, showing that it provides inaccurate estimates of intermediate state values and barely outperforms a random baseline when ranking alternative steps (see Section 7 for details).

- We propose VinePPO, introduced in Section 4, which takes advantage of the flexibility of language as an RL environment to compute unbiased value estimates, eliminating the need for large value networks and reducing memory requirements (up to 112GB for a 7B LLM).

---

[1]setting the Generalized Advantage Estimation (GAE, Schulman et al. 2016) parameter $\lambda = 1$

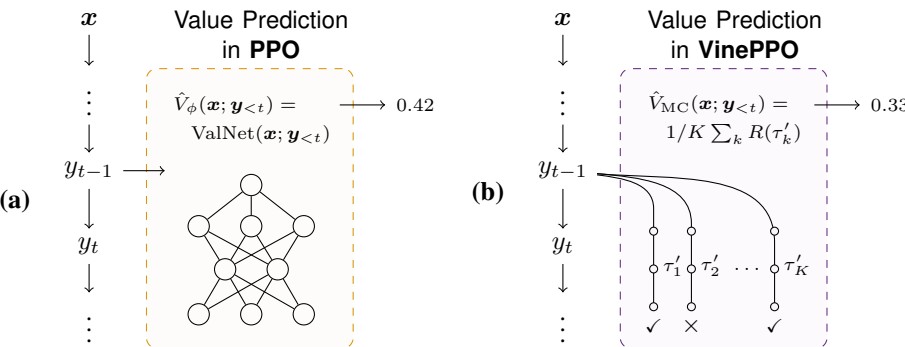

Figure 2: **(a)** PPO finetunes the model by adjusting action probabilities based on their advantage, which is primarily guided by the value network's value estimates. **(b)** VinePPO modifies standard PPO and obtains values estimates by simply *resetting* to intermediate states and using MC samples.

- VinePPO highlights the significance of credit assignment: It outperforms PPO and other baselines, especially on more challenging datasets. It achieves PPO's peak performance with fewer iterations (up to 9x), less wall-clock time (up to 3.0x), and better KL-divergence trade-off. See Section 6.

## 2 RELATED WORK

**Credit Assignment in Post-Training of LLM** PPO, as applied in RL from Human Feedback (RLHF, Ouyang et al. 2022), pioneered RL finetuning of LLMs. However, its computational over-head and hyperparameter sensitivity led to the development of simpler alternatives. RL-free methods such as DPO (Rafailov et al., 2023) operate in a bandit setting, treating the entire response as a single action. Similarly, rejection sampling methods like RestEM (Singh et al., 2024) finetune on full high-reward responses. RLOO (Ahmadian et al., 2024) and GRPO (Shao et al., 2024) abandon PPO's value network, instead using average reward from multiple samples as a baseline. Recent work has emphasized finer credit assignment, with Hwang et al. (2024) and Setlur et al. (2024) introducing MC-based methods to detect key errors in reasoning chains for use as ad-hoc mechanisms in DPO. Our work, by contrast, fully embraces the RL training, with the target of unlocking PPO's poten-tial. Parallel efforts have also focused on building better verifiers and reward models for per-step feedback, with recent attempts to automate their data collection using MC rollouts (Ma et al., 2023; Uesato et al., 2022; Luo et al., 2024; Wang et al., 2024). Our method is orthogonal to these methods, operating within PPO-based training to optimize a *given* reward, instead of designing new ones.

**Value Estimation in RL and Monte Carlo Tree Search (MCTS)** Deep RL algorithms are typi-cally categorized into value-based and policy-based methods. Policy-based methods like PPO usu-ally employ critic networks for value prediction. An exception is the *"Vine"* variant of TRPO (Schulman et al., 2015), which uses MC samples for state value estimation. The authors, however, note that the Vine variant is limited to environments that allow intermediate state resets, rare in typical RL settings[2]. However, language generation – when formulated as RL environment – en-ables such intermediate reset capabilities. In domains with similar reset capabilities, such as Go and Chess, MC-heavy methods like AlphaGo (Silver et al., 2016) and AlphaZero (Silver et al., 2017) have emerged. AlphaGo's architecture includes a policy, trained using expert moves and self-play, and a value network that predicts game outcomes. At inference, it employs tree search guided by MC rollouts and value network to select optimal moves. AlphaZero advances this approach by distilling MCTS outcomes into the policy. Recent works have adapted AlphaZero's principles to LLMs, em-ploying similar search techniques during inference to improve responses and during training to find better trajectories for distillation (Xie et al., 2024; Chen et al., 2024; Wan et al., 2024; Zhang et al., 2024; Hao et al., 2023). While this is a promising direction, our method is not an MCTS approach; it uses MC samples solely for value estimation during PPO *training* to improve credit assignment.

---

[2]This is reflected in the design of Gym (Towers et al., 2024), which only allows resets to the initial state.

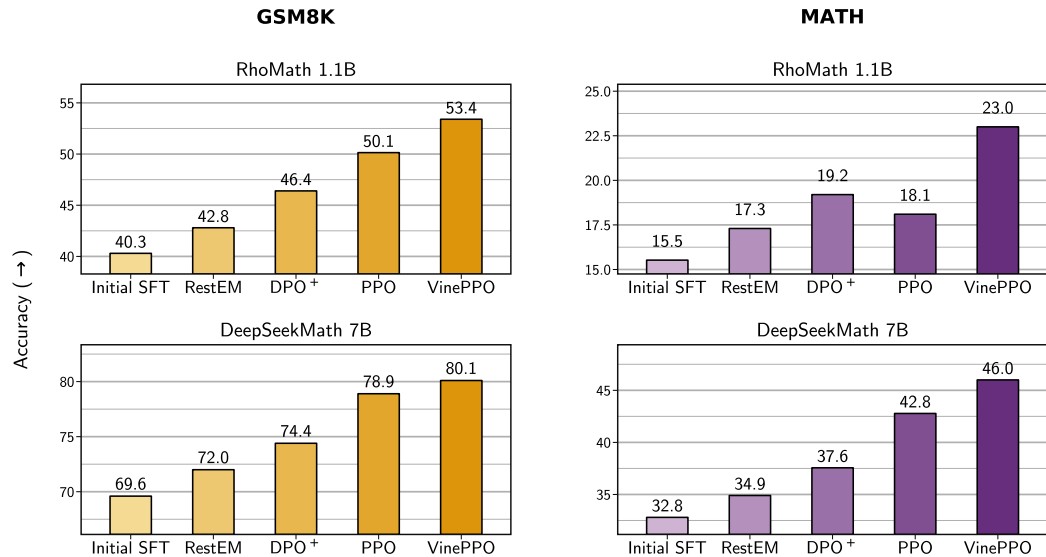

Figure 3: VinePPO outperforms standard PPO and other RL-free baselines on Pass@1 performance on MATH and GSM8K datasets, while also exhibiting scalability across different model sizes.

## 3 BACKGROUND

We focus on the RL tuning phase in the RLHF pipeline, following Ouyang et al. (2022); Shao et al. (2024). In this section, we provide an overview of actor-critic finetuning as implemented in PPO.

**RL Finetuning**   In this setup, the policy $\pi_\theta$ represents a language model that generates a response $\boldsymbol{y} = [y_0, \ldots, y_{T-1}]$ autoregressively given an input $\boldsymbol{x} = [x_0, \ldots, x_{M-1}]$. The goal of RL finetuning is to maximize the expected undiscounted ($\gamma = 1$) finite-horizon return, while incorporating a KL-divergence constraint to regularize the policy and prevent it from deviating too far from a reference policy $\pi_{\mathrm{ref}}$ (typically the initial supervised finetuned, SFT, model). The objective can be written as:

$$J(\theta) = \mathbb{E}_{\boldsymbol{x} \sim \mathcal{D}, \boldsymbol{y} \sim \pi(\cdot|\boldsymbol{x})} \left[ \mathcal{R}(\boldsymbol{x}; \boldsymbol{y}) \right] - \beta \, \mathrm{KL}[\pi_\theta \| \pi_{\mathrm{ref}}], \tag{1}$$

where $\mathcal{D}$ is the dataset of prompts, $\mathcal{R}(\boldsymbol{x}; \boldsymbol{y})$ is the complete sequence-level reward function, and $\beta$ controls the strength of the KL penalty. Note that the policy $\pi_\theta$ is initialized from $\pi_{\mathrm{ref}}$.

**Language Environment as an MDP**   Language generation is typically modeled as a token-level Markov Decision Process (MDP) in an actor-critic setting, where each response $\boldsymbol{y}$ is an episode. The state at time step $t$, $s_t \in \mathcal{S}$, is the concatenation of the input prompt and the tokens generated up to that point: $s_t = \boldsymbol{x}; \boldsymbol{y}_{<t} = [x_0, \ldots, x_{M-1}, y_0, \ldots, y_{t-1}]$. At each time step, the action $a_t$ corresponds to generating the next token $y_t$ from fixed vocabulary. The process begins with the initial state $s_0 = \boldsymbol{x}$, and after each action, the environment transitions to the next state, $s_{t+1} = s_t; [a_t]$, by appending the action $a_t$ to the current state $s_t$. In this case, since states are always constructed by concatenating tokens, the environment dynamics are known and the transition function is *deterministic*, i.e., $P(s_{t+1}|s_t, a_t) = 1$. During the generation process, the reward $r_t$ is set to zero for all intermediate actions $a_t$'s, with the sequence-level reward $\mathcal{R}(\boldsymbol{x}; \boldsymbol{y})$ only applied at the final step when the model stops generating. A trajectory $\tau = (s_0, a_0, s_1, a_1, \ldots)$ is therefore a sequence of state-action pairs, starting from the input prompt until the terminal state. Finally, we define the cumulative return of a trajectory $\tau$ as $R(\tau) = \sum_{t=0}^{T-1} r_t = r_{T-1} = \mathcal{R}(\boldsymbol{x}; \boldsymbol{y})$.

**Policy Gradient**   Given this MDP formulation, policy gradient methods like PPO maximize Equation 1 by repeatedly sampling trajectories and taking a step in the direction of the gradient $\boldsymbol{g}_{\mathrm{pg}} := \nabla_\theta J(\theta)$ to update the parameters. Policy gradient $\boldsymbol{g}_{\mathrm{pg}}$ takes the following form:

$$\boldsymbol{g}_{\mathrm{pg}} = \mathbb{E}_{\tau \sim \pi_\theta} \left[ \sum_{t=0}^{T-1} \nabla_\theta \log \pi_\theta(a_t|s_t) A(s_t, a_t) \right], \quad \text{where } s_t = \boldsymbol{x}; \boldsymbol{y}_{<t}, \ a_t = y_t, \tag{2}$$

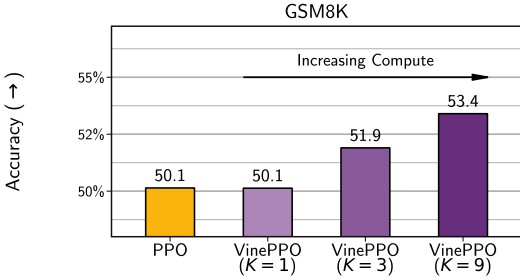 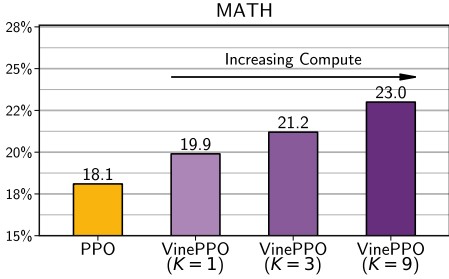

Figure 4: Impact of number of sampled trajectories $K$ for estimating $\hat{V}_{\mathrm{MC}}(s_t)$, evaluated on RhoMath 1.1B models. Increasing the number of rollouts improves task performance consistently.

where $A(s_t, a_t)$ is the *advantage* function. If $A(s_t, a_t) > 0$, $\boldsymbol{g}_{\mathrm{pg}}$ will increase the probability of action $a_t$ in state $s_t$, and decrease it when $A(s_t, a_t) < 0$. Intuitively, the advantage function quantifies how much better action $a_t$ is compared to average actions taken in state $s_t$ under the policy. Formally, it is defined as:

$$A(s_t, a_t) = Q(s_t, a_t) - V(s_t) = r_t + \gamma V(s_{t+1}) - V(s_t), \tag{3}$$

where $Q(s_t, a_t)$ is the state-action value and $V(s_t)$ is the per-state value function[3]. The value function, $V(s_t) : \mathcal{S} \to \mathbb{R}$, offers a long-term assessment of how desirable a particular state is under the current policy. Formally, it represents the expected cumulative reward obtained from starting in state $s_t$ and following the policy thereafter[4]: $V(s_t) = \mathbb{E}_{\tau \sim \pi_\theta} [R(\tau) \mid s_0 = s_t]$. PPO uses the same advantage-weighted policy gradient as in Equation 2, but constrains policy updates through clipping to ensure stable training. For full details, see Appendix A.

**Estimating Advantage via Value Networks** In practice, the advantage $A(s_t, a_t)$ is not known beforehand and is typically estimated by first using a value network $\hat{V}_\phi$ to approximate the *true value function* $V(s_t)$, then substituting the learned values into Equation 3 or alternative methods like GAE (Schulman et al., 2016). The value network is parameterized by $\phi$ and trained alongside the policy network $\pi_\theta$. The training objective for the value network minimizes the mean squared error between the predicted value and the empirical return:

$$\mathcal{L}_V(\phi) = \mathbb{E}_{\tau \sim \pi_\theta} \left[ \frac{1}{T} \sum_t \frac{1}{2} (\hat{V}_\phi(s_t) - G_t)^2 \right], \tag{4}$$

where $G_t = \sum_{t'=t}^{T-1} r_{t'}$ is the empirical return from state $s_t$. PPO uses the same objective for $\hat{V}_\phi$ but enhances stability by applying clipping during training (see Appendix A.1 for details). In RL-tuning of LLMs, the value network is often initialized using the initial SFT policy $\pi_{\mathrm{ref}}$ (or the reward model when available), with the language modeling head swapped out for a scalar head to predict values (Zheng et al., 2023). This setup leverages the prior knowledge of the pretrained model.

## 4 ACCURATE CREDIT ASSIGNMENT WITH VINEPPO

As outlined in Section 3, a step in the PPO gradient update aims to increase the probability of better-than-average actions while decreasing the probability of those that perform worse—a process quantified by the advantage $A(s_t, a_t)$. However, the true advantage is generally unknown and must be estimated, typically by substituting estimates from a value network into Equation 3. As we will elaborate in Section 7, value networks are often inaccurate and result in biased value computation. Fortunately, the language environment as an MDP (Section 3) offers a useful property that allows for unbiased estimation of $V(s_t)$. Since states are simply concatenated tokens, we can prompt the language model $\pi_\theta$ to generate continuations from any intermediate state. This flexibility allows

---

[3]Such derivation is possible as the language environment is deterministic.
[4]We drop the dependency on $\pi_\theta$ for brevity.

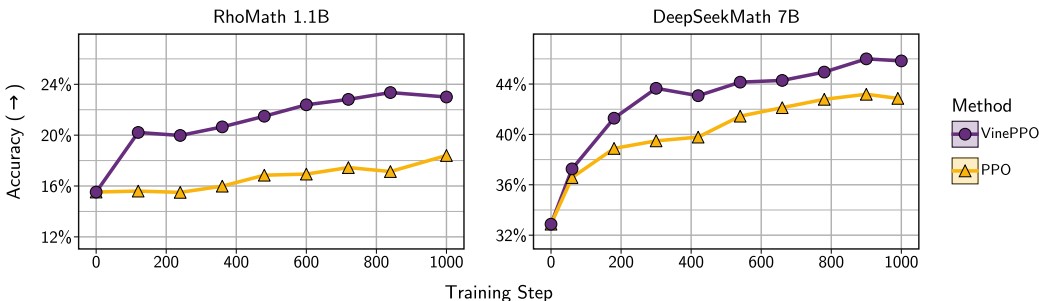

Figure 5: Comparison of the training behavior between VinePPO and PPO. VinePPO demonstrates consistently higher accuracy (as measured on the test set of MATH dataset) throughout the training. Refer to Appendix D for more detailed plots.

us to explore alternative future paths from arbitrary points in a generation. Moreover, recent advancements in LLM inference engines (Kwon et al., 2023; Zheng et al., 2024) have dramatically increased the speed of on-the-fly response generation[5]. This computational efficiency makes it feasible to conduct fast environment simulation, opening up unique opportunities for RL training of LLMs. VinePPO uses this property and estimates advantage via MC sampling. It only modifies the way advantages are estimated, leaving the rest of the standard PPO pipeline intact (Figure 2).

We start by estimating the true value $V(s_t)$. Instead of relying on a value network, for any intermediate state $s_t$, we sample $K$ independent trajectories $\tau^k$'s. The average return across these trajectories serves as the value estimate:

$$\hat{V}_{\text{MC}}(s_t) := \frac{1}{K} \sum_{k=1}^{K} R(\tau^k), \quad \text{where } \tau^1, \dots, \tau^K \sim \pi_\theta(\cdot \mid s_t). \tag{5}$$

This is a MC estimate of $V(s_t) = \mathbb{E}\left[R(\tau) \mid s_0 = s_t\right]$. Note that these trajectories are not trained on. Once the value $\hat{V}_{\text{MC}}(s_t)$ is computed, we estimate the advantages of each action using Equation 3:

$$\hat{A}_{\text{MC}}(s_t, a_t) := r(s_t, a_t) + \gamma \hat{V}_{\text{MC}}(s_{t+1}) - \hat{V}_{\text{MC}}(s_t). \tag{6}$$

For any $K \geq 1$, the policy gradient computed using the advantage estimator $\hat{A}_{\text{MC}}$ is an unbiased estimate of the gradient of expected return $\boldsymbol{g}_{\text{pg}}$. To enhance the efficiency of $\hat{A}_{\text{MC}}$, we group states within a reasoning step and compute a single advantage, which is assigned to all tokens in that step (examples in Appendix B). This trades off granularity for efficiency, allowing finer resolution with more compute, or coarser estimates with limited resources. The parameter $K$ also offers another trade-off between computational cost (i.e. more MC samples per state) and the variance of the estimator. As shown in Section 6.1, even $K = 1$ performs well.

In essence, VinePPO is a straightforward modification to the PPO pipeline, altering only the advantage computation. This minimal adjustment allows us to leverage PPO's benefits while enabling a systematic evaluation of the effect of unbiased advantage estimation and improved credit assignment. In the following sections, we compare various aspects such as task performance, computational efficiency, KL divergence, and robustness to shed light on the nature of these approaches.

## 5 EXPERIMENTAL SETUP

**Datasets and Pretrained LLMs**   We conduct our experiments using LLMs that show strong performance on mathematical reasoning: DeepSeekMath 7B (Shao et al., 2024) and RhoMath 1.1B (Lin et al., 2024), both of which have been trained on diverse mathematical and natural language corpora. Having different sized models allows evaluating the effect of scaling. We focus on mathematical reasoning datasets MATH (Hendrycks et al., 2021), consisting of *competition-level* mathematical problems, and GSM8K (Cobbe et al., 2021), containing simpler *grade-school level* math

---

[5]up to 5K tokens/second on a single Nvidia A100 GPU for a 7B LLM loaded in bfloat16.

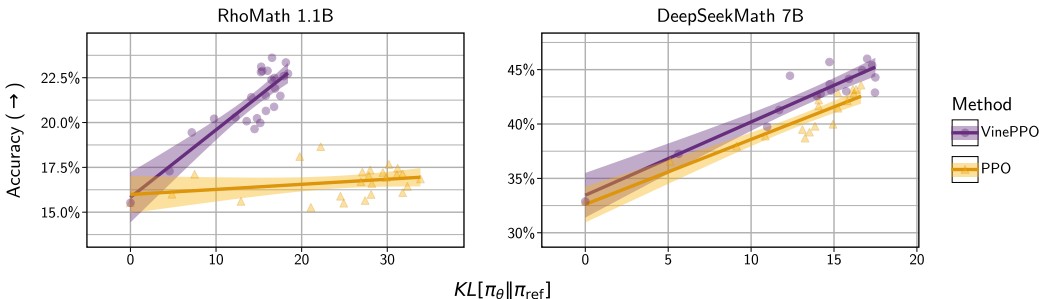

Figure 6: Task accuracy as a function of KL divergence during training on the MATH dataset. VinePPO achieves higher accuracy, reflecting more efficient credit assignment and focused updates.

word problems. Both datasets are well-established and present a range of difficulty levels that allow for comprehensive evaluation. For each dataset, we finetune the base LLMs on its respective training sets to obtain the initial SFT policy ($\pi_{\text{ref}}$). In all experiments, we employ *full-parameter finetuning* to allow utilization of models' full capacity (Sun et al., 2023; Biderman et al., 2024).

**Evaluation**  We evaluate model performance on the test sets of each dataset, using accuracy (Pass@1) as our primary metric, which measures the correctness of the final answers produced by the models. As our baseline, we adopt the standard PPO framework, as commonly implemented for LLM finetuning (Ouyang et al., 2022; Huang et al., 2024). Additionally, we compare them against RL-free methods that doesn't have explicit credit assignment mechanisms: RestEM (Singh et al., 2024), a form of Iterative Rejection Finetuning (Yuan et al., 2023; Anthony et al., 2017) and DPO$^+$ (Pal et al., 2024), variant of DPO with strong performance on reasoning tasks. All methods are initialized from the same SFT checkpoint to ensure a fair comparison.

**Training Details and Hyperparameters**  To ensure standard PPO (and its value network) has a healthy training and our evaluation reflects its full potential, we first focus our hyperparameter search on PPO parameters (such as KL penalty coefficient, batch size, minibatch size, GAE $\lambda$, number of epochs per iteration) and apply all well-known techniques and best practices (Huang et al., 2024; Ivison et al., 2024) in PPO tuning (Refer to Appendix C.2 for the full list). Following previous work (Pal et al., 2024; Singh et al., 2024), we set the task reward $\mathcal{R}$ to be a binary function that only checks final answer against the ground truth. VinePPO borrows the exact same hyperparameters from PPO and only modifies the advantage $A(s_t, a_t)$ estimation, keeping the rest of the pipeline unchanged. This allows us to isolate the effect of accurate credit assignment. We found that sampling $K = 9$ trajectories in $\hat{V}_{\text{MC}}$ performs well; the effect of varying $K$ is fully analyzed in Section 6.1. For the other baseline, we closely follow the original setup while ensuring consistency in training conditions for a fair comparison. We choose the best checkpoint based on a held-out validation set for all experiments. Full implementation details, including all hyperparameters and training procedures, are provided in Appendix C.6.

## 6 RESULTS

We evaluate the effect of accurate credit assignment on four key measures of model finetuning efficiency and success: task performance, KL divergence, temperature tolerance, and computational efficiency. Our experimental setup is designed to control for and isolate the impact of credit assignment on each of these measures.

### 6.1 TASK PERFORMANCE

VinePPO consistently outperforms standard PPO throughout training (Figure 5) and other baselines (Figure 3). More importantly, its performance gap widens in MATH which is a much more challenging reasoning task. Unlike VinePPO and PPO, DPO$^+$ and RestEM lacks any explicit mechanisms for credit assignment, opting instead to finetune the model on the full trajectory. Our experiments

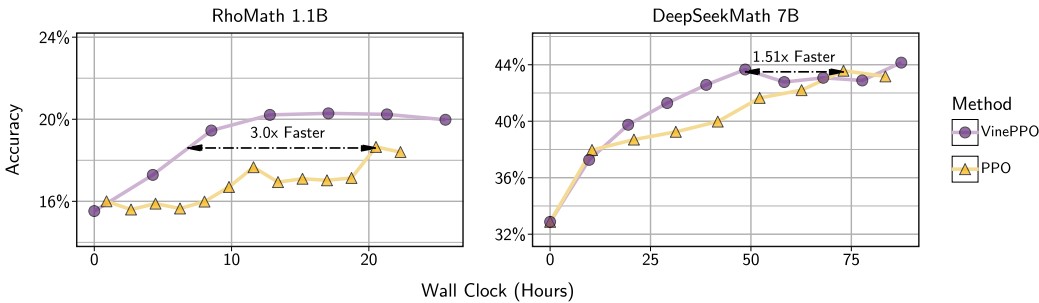

Figure 7: Accuracy vs. Wall Clock Time for both methods measured on the same hardware (shown only up to PPO's final performance). Despite VinePPO taking longer per iteration (up to 2x for 7B and 5x for 1.1B models), it passes PPO's peak performance in fewer iterations and less overall time.

show that these RL-free methods lags behind both PPO-based methods. For RestEM, the absence of targeted credit assignments likely leads to overfitting (Appendix C.5).

To assess the impact of $K$, the number of MC samples used to estimate the value, we run an ablation on RhoMath 1.1B, varying $K$ from 1 to 3 and then to 9. As shown in Figure 4, VinePPO demonstrates improved performance with higher $K$ values, as more MC samples reduce the variance of the $\hat{A}_{\mathrm{MC}}$ estimator. Notably, increasing $K$ provides a reliable approach to leveraging additional computational resources for better performance.

## 6.2 KL DIVERGENCE

The RL objective (Equation 1) balances maximizing task performance while constraining deviations from the initial policy $\pi_{\mathrm{ref}}$, measured by KL divergence. We analyze how VinePPO and PPO navigate this trade-off by plotting task accuracy against KL divergence $\mathrm{KL}[\pi_\theta \| \pi_{\mathrm{ref}}]$ throughout training (Figure 6). Results show VinePPO consistently achieves higher accuracy at same KL divergence, indicating more efficient use of the "KL budget." This efficiency stems from VinePPO's more precise credit assignment. As shown in Figure 1, many advantages are zero, and VinePPO excludes these steps from the loss. By avoiding unnecessary updates on non-contributing tokens, VinePPO reduces non-essential parameter adjustments that would inflate KL. See Appendix D.1 for full results.

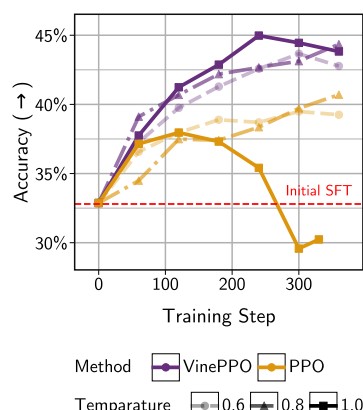

Figure 8: Test set accuracy during training with higher temperature presented for DeepSeekMath 7B and MATH dataset. VinePPO can tolerate higher temperatures.

## 6.3 TEMPERATURE TOLERANCE

Sampling temperature is a critical hyperparameter controlling the randomness of sampled trajectories. At higher temperatures models generates more diverse trajectories, accelerating early training through increased exploration. However, this diversity challenges PPO's value network, requiring generalization over a wider range of states. We compared VinePPO and PPO using temperatures $T \in \{0.6, 0.8, 1.0\}$ over the initial third of training steps. Figure 8 shows VinePPO consistently benefits from higher temperatures, achieving faster convergence. Conversely, PPO fails to benefit from increased exploration and even diverges at $T = 1.0$, where trajectories are most diverse.

## 6.4 COMPUTATIONAL EFFICIENCY

VinePPO and PPO require different resources: PPO uses a separate value network, requiring two times more GPU memory (up to 112GB for a 7B LLM, considering both model and its optimizer); VinePPO, conversely, relies on MC samples. This skips value network's memory requirements, but shifts the computational burden to increased LLM inferences, making VinePPO generally slower

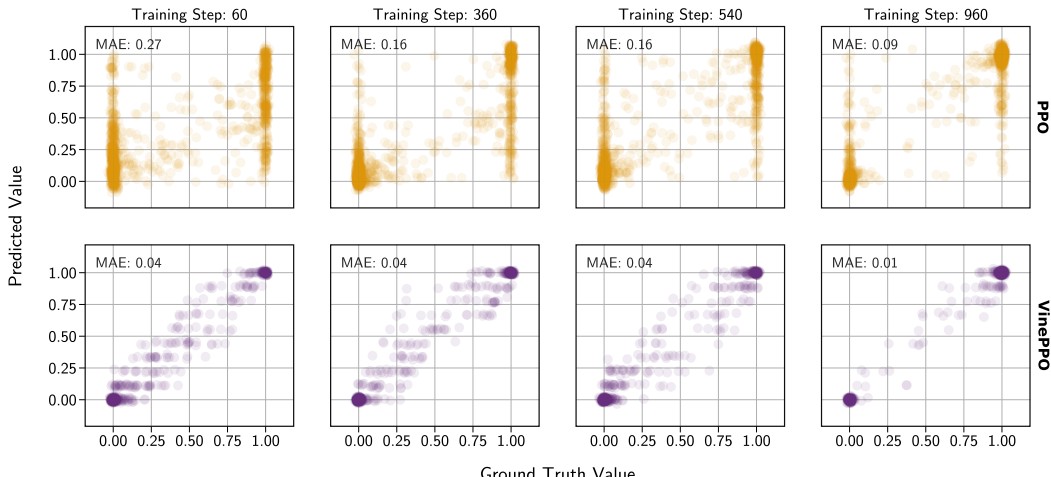

Figure 9: Distribution of predicted values for each state vs. ground truth (computed using 256 MC samples) during training for DeepSeekMath 7B on MATH dataset, highlighting the nature of errors. VinePPO achieves much lower Mean Absolute Error (MAE).

per iteration (up to 5x for RhoMath 1.1B and 2x for DeepSeekMath 7B). However, the effect of VinePPO's accurate credit assignment is substantial. Although slower per iteration, VinePPO achieves PPO's peak accuracy in *fewer gradient steps* and *less wall-clock time.* Figure 7 shows RhoMath 1.1B and DeepSeekMath 7B require about 3.0x and 1.51x less time and 9x and 2.8x fewer steps. This improvement occurs despite all hyperparameters being tuned for PPO. Therefore, switching to VinePPO offers a way to enhance performance within the same compute budget and serves as the only option when memory is constrained.

## 7 VALUE PREDICTION ANALYSIS

In this section, we explore the underlying reasons for the performance gap between PPO and VinePPO by closely analyzing the value prediction of both methods. First, we establish a *"ground truth"* value at each reasoning step within trajectories by running many MC samples (256 in our case) and averaging the returns. This provides a low-variance reference value. We then compare the value predictions in both methods against this ground truth. We present the results for DeepSeek-Math 7B on the MATH dataset (full analysis with other models and datasets in Appendix D.2).

**Accuracy** Figure 9 presents the distribution of value predictions at each reasoning step. The errors produced by VinePPO and PPO differ significantly. VinePPO's estimates are unbiased, with variance peaking at 0.5 and dropping to zero at 0 and 1. PPO's value network shows high bias, often misclassifying bad states (ground truth near 0) as good and vice versa. To further visualize accuracy, we classify a value prediction as "correct" if it falls within 0.05 of the ground truth. The accuracy of this formulation is shown in Figure 11.a. PPO's value network starts with low accuracy, gradually improving to 65%. VinePPO, however, consistently achieves 70-90% accuracy throughout training.

**Top Action Identification** In value-based RL, ranking actions correctly is more crucial than absolute value accuracy. While PPO, as a policy gradient method, requires accurate value estimates to compute meaningful advantages, it is still a compelling question whether PPO's value network, despite its bias, can maintain correct action ranking. To investigate, we sample five new next steps from the same initial state and evaluate if the method correctly identifies the resulting next state with the highest ground truth value. As shown in Figure 11.b, PPO's value network performs near chance levels for much of the training, with slight improvements over time. In contrast, VinePPO consistently identifies the top action with high accuracy throughout training.

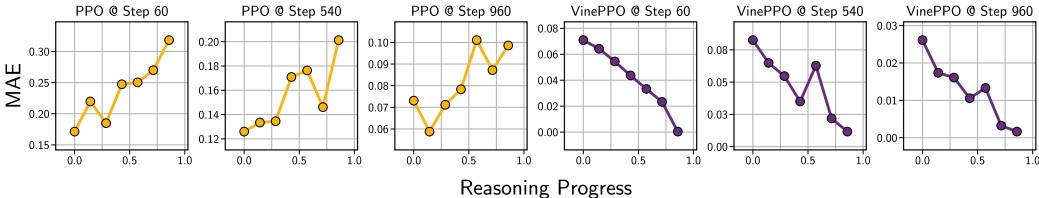

Figure 10: Visualizing the Mean Absolute Error (MAE) of the value predictions at different point of the reasoning chain. Value Network in PPO fails to generalize as the reasoning chain progresses, while VinePPO's value estimates become more accurate as the model become more deterministic.

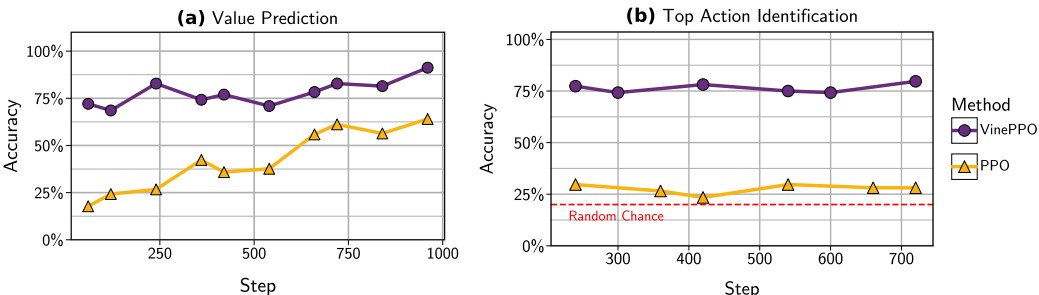

Figure 11: **(a)** Value prediction accuracy formulated as a classification problem, where a prediction is considered correct if it falls within 0.05 of the ground truth. **(b)** Accuracy of identifying the top action in a set of five possible next states. VinePPO consistently outperforms the value network.

**Error Per Reasoning Step**  To understand value computation mechanisms, we visualize the prediction error at each reasoning step within a trajectory. As shown in Figure 10, PPO's estimation error increases as reasoning progresses. We hypothesize this occurs because early steps have lower diversity and resemble training data more, allowing the value network to rely on memorization. Later, as space of states become much larger, they become unfamiliar and the network struggles to generalize. VinePPO's prediction error decreases with reasoning progression. We attribute this to the model becoming more deterministic in later steps as it conditions on bigger and longer context. This determinism enables more accurate estimates from the same number of MC samples.

# 8 DISCUSSION

Accurate credit assignment has profound implications on the performance of RL tuning of LLMs. As we've demonstrated, standard PPO, despite outperforming most RL-free baselines, suffers from suboptimal value estimation. More importantly, its scaling behavior is concerning; PPO struggles with increasingly diverse trajectories and tends to perform worse as tasks become more complex.

VinePPO, on the other hand, is a viable alternative. As shown in Section 6.4, it offers lowered memory requirements and better performance with the same computational budget. VinePPO could also be a particularly attractive option for frontier LLMs as even doubling the post-training compute is negligible compared to their pre-training costs (Ouyang et al., 2022)[6]. Given the major investments in pre-training compute and data collection of these models, it is imperative for model developers to employ post-training methods that provide more accurate updates, avoiding the high-variance adjustments caused by inferior credit assignment. Additionally, VinePPO offers a straightforward scaling axis: increasing the number of MC samples directly enhances performance with additional compute. Unlike recent approaches that focus on increasing inference-time compute to boost performance (OpenAI, 2024; Bansal et al., 2024), VinePPO's training compute is amortized over all future inferences. Note that the computational workload of VinePPO is highly parallelizable with linear scalability, making it well-suited for large-scale training.

---

[6]For example, InstructGPT used nearly 60 times more compute for pre-training (Ouyang et al., 2022).

The unique properties of the language environment are what enabled VinePPO to be viable credit assignment option; it may have limited practical use in traditional deep RL policy gradient methods. This suggests that adapting RL techniques to LLMs requires careful consideration and perhaps a reevaluation of underlying assumptions. Overall, our work highlights the potential of well-tuned RL finetuning strategies with proper credit assignment, and we hope it encourages further research into optimizing RL post-training pipelines for LLMs.

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

## A  REVIEWING PPO

PPO, as used in RL tuning of LLMs, formulates language generation as token-level MDP (Section 3), where each response $\boldsymbol{y}$ is an episode. The state at time step $t$, $s_t \in \mathcal{S}$, is the concatenation of the prompt and the tokens generated so far: $s_t = \boldsymbol{x}; \boldsymbol{y}_{<t} = [x_0, \ldots, x_{M-1}, y_0, \ldots, y_{t-1}]$. The action $a_t$ corresponds to generating the next token $y_t$ from the model's vocabulary. Given a prompt $\boldsymbol{x}$, an episode of this MDP starts from the initial state $s_0 = \boldsymbol{x}$, and with each action taken, the environment moves to a subsequent state, $s_{t+1} = s_t; [a_t]$, by adding the action $a_t$ to the existing state $s_t$. In the language environment, because states are always formed by concatenating tokens, the environment dynamics are fully known, and the transition function is *deterministic*, meaning $P(s_{t+1}|s_t, a_t) = 1$. Throughout the generation process, the reward $r_t$ is set to zero for all intermediate actions $a_t$, with the sequence-level reward $\mathcal{R}(\boldsymbol{x}; \boldsymbol{y})$ applied only at the final step when the model stops the generation. That is:

$$r_t = r(s_t, a_t) = \begin{cases} \mathcal{R}(\boldsymbol{x}; \boldsymbol{y}) & \text{if } t = T-1, \text{ where } s_{t+1} = \boldsymbol{y} \text{ is terminal,} \\ 0 & \text{otherwise.} \end{cases} \quad (7)$$

A trajectory $\tau = (s_0, a_0, s_1, a_1, \ldots)$ thus represents a sequence of state-action pairs that begins at the input prompt and continues until reaching the terminal state. Finally, the cumulative return of a trajectory $\tau$ is defined as $R(\tau) = \sum_{t=0}^{T-1} r_t = r_{T-1} = \mathcal{R}(\boldsymbol{x}; \boldsymbol{y})$.

The goal of RL tuning is to maximize the expected return of the model's responses to prompts in the dataset, as defined by the reward function $\mathcal{R}$ (Equation 1). PPO, similar to other policy gradient methods, achieves this goal by repeatedly sampling trajectories for a batch of prompt sampled from $\mathcal{D}$ and taking multiple optimization steps in the direction of the gradient $\boldsymbol{g}_{\text{ppo}}$ to update the parameters. PPO gradient $\boldsymbol{g}_{\text{ppo}}$ is defined as the gradient of the following loss:

$$\mathcal{L}_{\text{ppo}}(\theta) = \mathbb{E}_{\tau \sim \pi_{\theta_k}} \left[ \sum_{t=0}^{T-1} \min \left( \frac{\pi_\theta(a_t \mid s_t)}{\pi_{\theta_k}(a_t \mid s_t)} A_t^{\theta_k}, \ \text{clip}(\theta) A_t^{\theta_k} \right) - \beta \, \text{KL}[\pi_\theta \parallel \pi_{\text{ref}}] \right] \quad (8)$$

where $\pi_{\theta_k}$ is the policy at the previous iteration, $\epsilon$ is the clipping parameter, $\beta$ is the KL penalty coefficient, $A_t^{\theta_k} = A^{\theta_k}(s_t, a_t)$ is the advantage estimate for policy $\pi_{\theta_k}$, and the $\text{clip}(\theta)$ function is:

$$\text{clip}(\theta) = \text{clip} \left( \frac{\pi_\theta(a_t \mid s_t)}{\pi_{\theta_k}(a_t \mid s_t)}, 1 - \epsilon, 1 + \epsilon \right). \quad (9)$$

Note that the KL penalty could be also added to the reward function $\mathcal{R}$. We follow the more recent implementations (Shao et al., 2024; Qwen, 2024), where it is added to the loss function. The KL term can be computed using the following unbiased estimator (Schulman, 2020):

$$\hat{\text{KL}}(\theta) = \frac{\pi_{\text{ref}}(a_t \mid s_t)}{\pi_\theta(a_t \mid s_t)} - \log \frac{\pi_{\text{ref}}(a_t \mid s_t)}{\pi_\theta(a_t \mid s_t)} - 1, \quad (10)$$

where $\pi_{\text{ref}}$ denotes the reference model (initial SFT).

### A.1  VALUE NETWORK

In addition to the policy $\pi_\theta$, PPO also trains a separate value network $\hat{V}_\phi$ to obtain an estimate the true values $V(s_t)$ of states $s_t$. Parameterized by $\phi$, the value network is trained alongside the policy network $\pi_\theta$ using the following loss:

$$\mathcal{L}_{\text{ValNet}}(\phi) = \frac{1}{2} \mathbb{E}_{\tau \sim \pi_\theta} \left[ \frac{1}{T} \sum_{t=0}^{T-1} \max \left( \left\| \hat{V}_\phi(s_t) - G_t \right\|^2, \ \left\| \text{clip}(\phi) - G_t \right\|^2 \right) \right] \quad (11)$$

where $\hat{V}_{\phi_k}$ is the value network at the previous iteration, $G_t = \sum_{t'=t}^{T-1} \gamma^{t'-t} r_{t'}$ is the empirical return from state $s_t$, $\epsilon'$ is a value clipping parameter, and the $\text{clip}(\theta)$ is defined as:

$$\text{clip}(\phi) = \text{clip} \left( \hat{V}_\phi(s_t), \hat{V}_{\phi_k}(s_t) - \epsilon', \hat{V}_{\phi_k}(s_t) + \epsilon' \right). \quad (12)$$

In RL-tuning of LLMs, the value network is typically initialized from the initial policy $\pi_{\text{ref}}$ (or the reward model, if available), replacing the language modeling head with a scalar output head to predict values (Zheng et al., 2023) This approach takes advantage of the base model's prior knowledge for value estimation.

**Prompt**

If we express $3x^2 + x - 4$ in the form $a(x-h)^2 + k$, then what is $k$?

**Response**

We complete the square.[†] First, we factor 3 out of the terms $3x^2 + x$ to get $3\left(x^2 + \frac{x}{3}\right)$.[†] We can square $x + \frac{1}{6}$ to get $x^2 + \frac{x}{3} + \frac{1}{36}$, so[†]

$$3\left(x^2 + \frac{x}{3}\right)^† = 3\left[\left(x + \frac{1}{6}\right)^2 - \frac{1}{36}\right]^†$$

$$= 3\left(x + \frac{1}{6}\right)^2 - \frac{3}{36}^†$$

$$= 3\left(x + \frac{1}{6}\right)^2 - \frac{1}{12},^†$$

and

$$3\left(x^2 + \frac{x}{3}\right) - 4^† = 3\left(x + \frac{1}{6}\right)^2 - \frac{1}{12} - 4^†$$

$$= 3\left(x + \frac{1}{6}\right)^2 - \frac{49}{12}.$$

We see that $k =$[†] $\boxed{-\frac{49}{12}}$.[†]

**Prompt**

Suppose the roots of the polynomial $x^2 - mx + n$ are positive prime integers (not necessarily distinct). Given that $m < 20$, how many possible values of $n$ are there?

**Response**

Let $p$ and $q$ be the prime roots.[†] Then, we know that $m = p + q$ and $n = pq$.[†] Since $m < 20$, the primes $p$ and $q$ must both be less than 20.[†]

The primes less than 20 are 2, 3, 5, 7, 11, 13, 17, 19.[†] Now we list all possible pairs $(p, q)$ such that $p + q < 20$,[†] remembering to also include the cases in which $p = q$:[†]

$(2, 2), (2, 3), (2, 5), (2, 7), (2, 11), (2, 13), (2, 17)$[†]

$(3, 3), (3, 5), (3, 7), (3, 11), (3, 13)$[†]

$(5, 5), (5, 7), (5, 11), (5, 13)$[†]

$(7, 7), (7, 11)$

There are $7 + 5 + 4 + 2 = 18$ pairs in total.[†] Each pair produces a value for $n$, and furthermore,[†] these values are all distinct, because every positive integer has a unique prime factorization.[†] Thus, there are $\boxed{18}$ possible values for $n$.[†]

Figure B.1: Examples of solutions separated into its reasoning steps on the MATH dataset. Steps are highlighted using distinct colors. [†] denotes the reasoning step boundary.

**Advantage Estimation** Once the estimated values $\hat{V}_\phi(s_t)$ are obtained, the advantages $A(s_t, a_t)$ are computed using the GAE (Schulman et al., 2016):

$$A(s_t, a_t) \approx \hat{A}^{\text{GAE}}(s_t, a_t) \tag{13}$$

$$= (1 - \lambda)\left(\hat{A}_t^{(1)} + \lambda\hat{A}_t^{(2)} + \lambda^2\hat{A}_t^{(3)} + \dots\right) \tag{14}$$

$$= \sum_{l=0}^{\infty} (\gamma\lambda)^l \delta_{t+l} \tag{15}$$

$$= \sum_{l=0}^{\infty} (\gamma\lambda)^l \left(r_{t+l} + \gamma\hat{V}_\phi(s_{t+l+1}) - \hat{V}_\phi(s_{t+l})\right) \tag{16}$$

where $\delta_t = r_t + \gamma\hat{V}_\phi(s_{t+1}) - \hat{V}_\phi(s_t)$ is the temporal difference error, $\lambda$ is the GAE parameter, and $\gamma$ is the discount factor. Also, we have:

$$\hat{A}_t^{(k)} := \sum_{l=0}^{k-1} \gamma^l \delta_{t+l} = r_t + \gamma r_{t+1} + \dots + \gamma^{k-1} r_{t+k-1} + \gamma^k \hat{V}_\phi(s_{t+k}) - \hat{V}_\phi(s_t). \tag{17}$$

Adjusting the GAE parameter $\lambda$ allows for a trade-off between bias and variance in the advantage estimates. However, as we discuss in Appendix C.6, we found that $\lambda = 1$ works best in our experiments (similar to the findings of Trung et al. (2024) and Ahmadian et al. (2024)). In this case, the GAE simplifies to the following form (assuming $\gamma = 1$): $\hat{A}^{\text{GAE}}(s_t, a_t) = \sum_{t'=t}^{T-1} r_{t'} - \hat{V}_\phi(s_t)$.

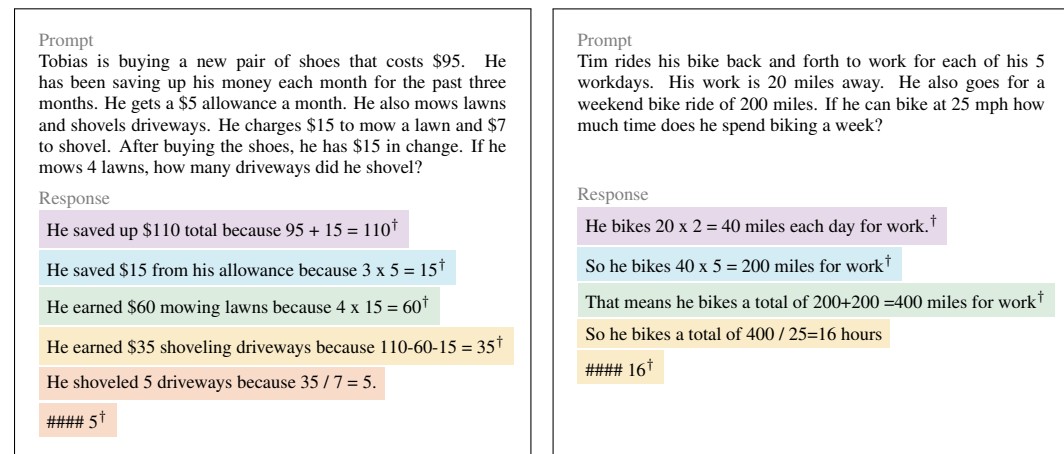

Figure B.2: Examples of solutions separated into its reasoning steps on the GSM8K dataset. Steps are highlighted using distinct colors. $^{\dagger}$ denotes the reasoning step boundary.

# B    REASONING STEP SEPARATION EXAMPLES

In this section, we outline the methodology used to segment solutions into discrete reasoning steps for the MATH and GSM8K datasets, as illustrated in Figures B.1 and B.2.

For the MATH dataset, we begin by splitting solutions based on clear natural boundaries such as newline characters or punctuation marks (e.g., periods or commas). Care is taken to avoid splitting within mathematical expressions, ensuring that mathematical formulas remain intact. After this initial segmentation, if any resulting step exceeds 100 characters, we further try to divide it by identifying logical breakpoints, such as equal signs (=) within math mode.

For the GSM8K dataset, we take a simpler approach, segmenting the reasoning steps by newlines alone as with this task newlines already serve as natural delimiters.

# C    EXPERIMENTAL DETAILS

## C.1    DATASETS

We focus on mathematical reasoning datasets that require step-by-step solutions and are widely used to evaluate the reasoning capabilities of LLMs. Below is a brief overview of the datasets used in our experiments:

**MATH (Hendrycks et al., 2021)**    The MATH dataset contains problems from high school math competitions, covering a wide range of topics such as algebra, geometry, and probability. For our experiments, we use the OpenAI split provided by Lightman et al. (2024), which consists of 500 problems for testing and 12,500 problems for training. We further divide the training set into 11,500 problems for training and 500 problems for validation. Each problem includes a step-by-step solution, ending in a final answer marked by \boxed{} in the solution (e.g., "*..so the smallest possible value of c is* $\boxed{\pi}$"). This marking allows for verification of the correctness of model-generated responses by comparing the final answer to the ground truth. We use the scripts provided by Lewkowycz et al. (2022), Lightman et al. (2024), and Shao et al. (2024) to extract and compare the final answers to the ground truth.

**GSM8K (Cobbe et al., 2021)**    The GSM8K dataset comprises high-quality grade-school math problems, requiring basic arithmetic or elementary algebra to solve. Although simpler than the MATH dataset, GSM8K is still widely used to assess the reasoning capabilities of LLMs. It contains 1,319 problems for testing and 7,473 for training. To create a validation set, we further split the training set into 7,100 problems for training and 373 for validation. Verifying the correctness of

Table 1: Summary of PPO hyperparamters used in the experiments.

| Parameter | Value | |
|---|---|---|
| TRAINING | | |
| Optimizer | AdamW | |
| Adam Parameters $(\beta_1, \beta_2)$ | (0.9, 0.999) | |
| Learning rate | $1 \times 10^{-6}$ | |
| Weight Decay | 0.0 | |
| Max Global Gradient Norm for Clipping | 1.0 | |
| Learning Rate Scheduler | Polynomial | |
| Warm Up | 3% of training steps | |
| # Train Steps For MATH dataset | 1000 steps (around 8 dataset epochs) | |
| # Train Steps For GSM8K dataset | 650 steps (around 8 dataset epochs) | |
| GENERAL | | |
| Maximum Response Length | 1024 tokens | |
| Maximum Sequence Length for RhoMath 1.1B | 2048 tokens | |
| Maximum Sequence Length for DeepSeekMath 7B | 2500 tokens | |
| PPO | | |
| # Responses per Prompt | 8 | Search Space: $\{8, 16, 32\}$ |
| # Episodes per PPO Step | 512 | Search Space: $\{256, 512\}$ |
| # Prompts per PPO Step | $512/8 = 64$ | |
| Mini-batch Size | 64 | |
| # Inner epochs per PPO Step | 2 | Search Space: $\{1, 2\}$ |
| Sampling Temperature | 0.6 | Search Space: $\{0.6, 0.8, 1.0\}$ |
| Discount Factor $\gamma$ | 1.0 | |
| GAE Parameter $\lambda$ | 1.0 | Search Space: $[0.95 - 1.0]$ |
| KL Penalty Coefficient $\beta$ | 1e-4 | Search Space: $\{1e-1, 1e-2, 3e-3, 1e-4\}$ |
| Policy Clipping Parameter $\epsilon$ | 0.2 | |
| Value Clipping Parameter $\epsilon'$ | 0.2 | |

Table 2: Summary of RestEM hyperparamters used in the experiments.

| Parameter | Value | |
|---|---|---|
| TRAINING | | |
| Optimizer | AdamW | |
| Adam Parameters $(\beta_1, \beta_2)$ | (0.9, 0.999) | |
| Learning rate | $1 \times 10^{-6}$ | |
| Weight Decay | 0.0 | |
| Max Global Gradient Norm for Clipping | 1.0 | |
| Learning Rate Scheduler | Polynomial | |
| Warm Up | 3% of training steps | |
| RESTEM | | |
| # iterations | 10 | |
| # Sampled Responses per Prompt | 8 | Search Space: $\{8, 32\}$ |
| Sampling Temperature | 0.6 | Search Space: $\{0.6, 0.8, 1.0\}$ |
| Checkpoints every # iteration | 500 step | |
| Checkpoint Selection | until validation improves | |
| | Search Space: {until validation improves, best validation} | |

model responses is straightforward, as the final answer is typically an integer, marked by #### in the solution.

## C.2 PPO IMPLEMENTATION

To ensure our PPO implementation is robust, and our evaluation reflects its full potential, we have applied a set of well-established techniques and best practices from the literature (Huang et al., 2024;

Table 3: Summary of DPO-Positive hyperparameters used in the experiments.

| Parameter | Value | |
|---|---|---|
| **TRAINING** | | |
| Optimizer | AdamW | |
| Adam Parameters $(\beta_1, \beta_2)$ | (0.9, 0.999) | |
| Learning rate | $1 \times 10^{-6}$ | |
| Weight Decay | 0.0 | |
| Max Global Gradient Norm for Clipping | 1.0 | |
| Learning Rate Scheduler | Polynomial | |
| Warm Up | 3% of training steps | |
| **DPO-POSITIVE** | | |
| # DPO-$\beta$ | 0.1 for MATH, 0.3 for GSM8K | |
| # DPO-Positive-$\lambda$ | 50. | |
| # Epochs | 3 | Search Space: $\{3, 8\}$ |
| # Sampled Responses per Prompt | 64 | Search Space: $\{8, 64\}$ |
| # Pairs per prompt | 64 | Search Space: $\{8, 64\}$ |
| Sampling Temperature | 0.6 | |

Ivison et al., 2024; Zheng et al., 2023). Below, we outline the key implementation details that were most effective in our experiments:

- **Advantage Normalization**: After calculating the advantages, we normalize them to have zero mean and unit variance, not only across the batch but also across data parallel ranks. This normalization step is applied consistently in both our PPO and VinePPOimplementations.

- **Reward Normalization**: We follow Ivison et al. (2024) and do not normalize the rewards, as the reward structure in our task is already well-defined within the range of $[0, 1]$. Specifically, correct responses are assigned a reward of $1$, while incorrect responses receive $0$.

- **End-of-Sequence (EOS) Trick**: As detailed in Appendix A, rewards are only applied at the final token of a response, which corresponds to the EOS token when the response is complete. For responses that exceed the maximum length, we truncate the response to the maximum length and apply the reward to the last token of the truncated sequence. We also experimented with penalizing truncated responses by assigning a negative reward (-1), but this did not lead to performance improvements.

- **Dropout Disabling**: During the RL tuning phase, we disable dropout across all models. This ensures that the log probabilities remain consistent between different forward passes, thereby avoiding stochastic effects that could hurt training stability.

- **Fixed KL Coefficient** We use a constant coefficient for the KL penalty. Although the original PPO implementation for finetining language models (Ziegler et al., 2019) utilized an adaptive KL controller, more recent implementations typically do not use this approach (Ouyang et al., 2022; Touvron et al., 2023; Xu et al., 2024).

## C.3 SFT MODELS

To ensure a systematic and reproducible evaluation, we create our SFT models $\pi_{\text{ref}}$ by finetuning the *base pretrained LLMs* (as opposed to their "Instruct" version) on the training splits of the respective datasets. Specifically, we produce four distinct SFT models: two base LLM (DeepSeekMath 7B and RhoMath 1.1B ) across two datasets (MATH and GSM8K). The base models are finetuned using the Adam optimizer without weight decay. We employ a learning rate warm-up over 6% of the total training steps. Each model is trained for three epochs with a batch size of 64, and the best checkpoint is selected based on validation accuracy. For each SFT model, we conduct a hyperparameter sweep over learning rates in the range $\{1 \times 10^{-7}, 3 \times 10^{-7}, 1 \times 10^{-6}, 3 \times 10^{-6}, 1 \times 10^{-5}, 3 \times 10^{-5}, 8 \times 10^{-5}, 1 \times 10^{-4}\}$ to ensure optimal performance. We then use these SFT models as the initial checkpoint for training the methods mentioned in our paper.

## C.4 EVALUATION

We evaluate each method's performance on the test sets of each dataset. For example, when we report that PPO achieves 42.8% accuracy on the MATH dataset for the DeepSeekMath 7B model, this means the PPO training was initialized with the SFT model specific to DeepSeekMath 7B on the MATH dataset, and accuracy was measured on the MATH test set. Our primary evaluation metric is accuracy, specifically $\text{Pass}@1$, which reflects the percentage of correctly answered problems on the first attempt. This metric is crucial because it represents a realistic user interaction, where the model is expected to deliver a correct answer without the need for multiple tries. For each evaluation, we sample a response from the model for a given prompt, using a maximum token length of 1024 and a temperature of 0.35. A response is considered correct if its final answer matches the ground truth final answer, as detailed in Appendix C.1. Furthermore, each accuracy score is averaged over 16 evaluation rounds, each conducted with different random seeds. This will ensure a robust and low variance assessment of model performance.

## C.5 BASELINES

**DPO$^+$ (DPO-Positive) (Pal et al., 2024)** The original DPO method has a failure mode when the edit distance between positive (correct) and negative (incorrect) responses is small. In these cases, the probability of both responses tends to decrease. This issue is especially common in reasoning and mathematical tasks, where multiple solution paths may involve similar equations or steps. Although DPO achieves its goal by reducing the probability of the incorrect response more than the correct one, it ultimately still lowers the likelihood of generating the correct response. This undermines model performance, making it a failure mode despite partially fulfilling the DPO objective. (Pal et al., 2024; Hwang et al., 2024). While previous methods mitigated this issue by maintaining a high edit distance between positive and negative response pairs, DPO-Positive (Pal et al., 2024) addresses it more effectively. It introduces an additional term to the DPO objective, penalizing any reduction in the probability of the correct response below its probability under the reference model. This ensures that the correct response is not overly suppressed, even when the edit distance is small. The final objective of DPO-Positive is::

$$\mathcal{L}_{\text{DPO-Positive}}(\pi_\theta; \pi_{\text{ref}}) = -\mathbb{E}_{(x,y_w,y_l) \sim \mathcal{D}} \left[ \log \sigma \left( \beta \underbrace{\left( \log \frac{\pi_\theta(y_w|x)}{\pi_{\text{ref}}(y_w|x)} - \log \frac{\pi_\theta(y_l|x)}{\pi_{\text{ref}}(y_l|x)} \right)}_{\text{DPO Original term}} \right. \right.$$

$$\left. \left. - \lambda \cdot \underbrace{\max \left( 0, \log \frac{\pi_{\text{ref}}(y_w|x)}{\pi_\theta(y_w|x)} \right)}_{\text{DPO-Positive additional term}} \right) \right] \tag{18}$$

where $\lambda$ is a hyperparameter controlling the weight of the additional term keeping the probabilities of correct responses high. We chose DPO-Positive as a baseline due to its strong performance in (Setlur et al., 2024).

**RestEM (Singh et al., 2024)** RestEM is an iterative method where, in each iteration, the base model is trained on correct, self-generated responses from the chosen checkpoint of the previous iteration. RestEM takes gradient steps to maximize this objective until the fine-tuned model's accuracy drops on a validation split. The objective of the fine-tuning process is to maximize the log-likelihood of correct responses. Training the model with a maximum likelihood objective on correct responses is mathematically equivalent to training the model with REINFORCE (Sutton et al., 1999), without a baseline, where the entire response is treated as a single action. The reward is 1 when the response is correct, and 0 otherwise. Specifically, we have:

$$\underbrace{\mathbb{E}_{\boldsymbol{x} \sim \mathcal{D}, \boldsymbol{y} \sim \pi(\cdot|\boldsymbol{x}), \mathcal{R}(\boldsymbol{x};\boldsymbol{y})=1} \left[ \nabla_\theta \log P_\theta(\boldsymbol{y}|\boldsymbol{x}) \right]}_{\text{max log-likelihood on correct responses}} = \underbrace{\mathbb{E}_{\boldsymbol{x} \sim \mathcal{D}, \boldsymbol{y} \sim \pi(\cdot|\boldsymbol{x})} \left[ \nabla_\theta \log P_\theta(\boldsymbol{y}|\boldsymbol{x}) \mathcal{R}(\boldsymbol{x};\boldsymbol{y}) \right]}_{\text{REINFORCE}} \tag{19}$$

Therefore, maximizing log-likelihood training on correct responses is equivalent to train with policy gradient without precise credit assignment, such as without advantages for specific actions. In our experiments, we observe the impact of this limitation in both Figure C.3 and Figure C.4 where RestEM overfits on the training data.

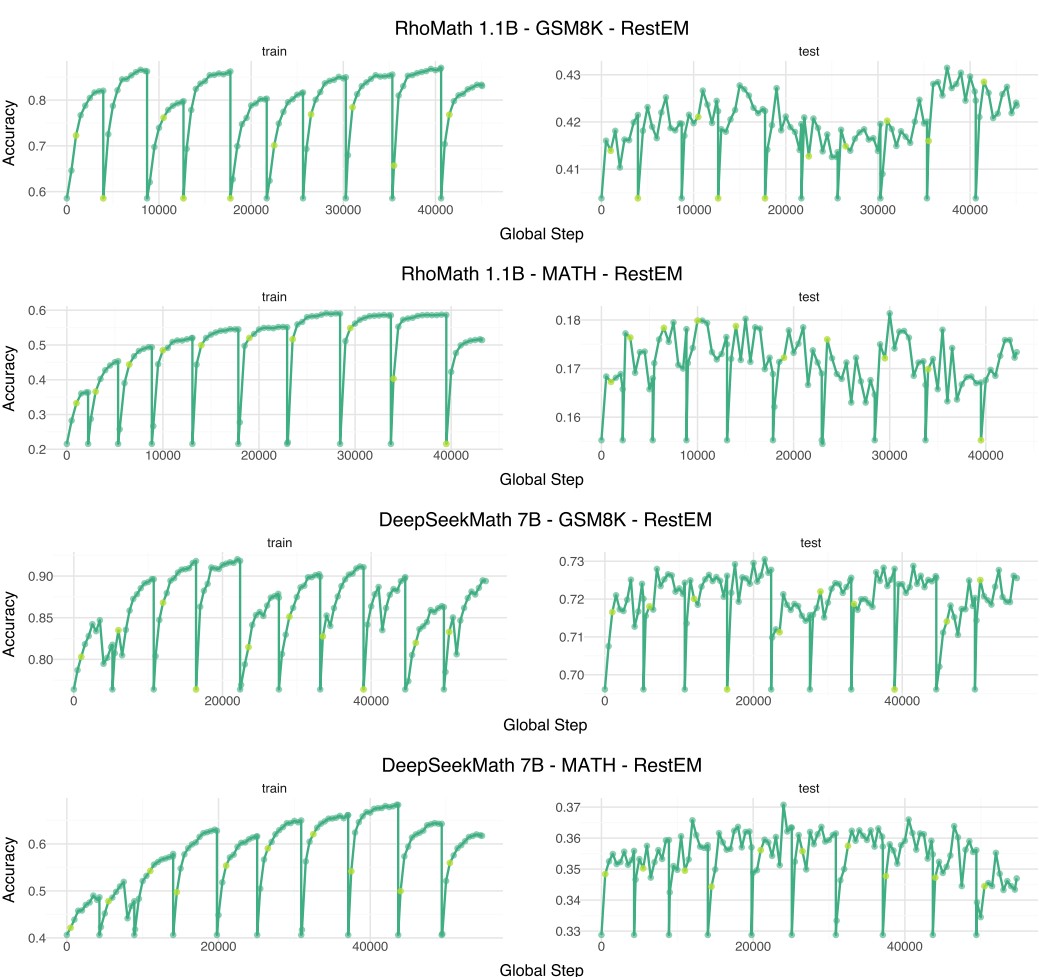

Figure C.3: Performance comparisons across different models and datasets: (a) RhoMath 1.1B on GSM8K, (b) RhoMath 1.1B on MATH, (c) DeepSeekMath 7B on GSM8K, and (d) DeepSeekMath 7B on MATH. The yellow points are chosen checkpoints based on the RestEM rule. Within each iteration, we train on the generated data of the chosen checkpoint for eight epochs and then we choose the first place where performance on a validation split drops following Singh et al. (2024)

## C.6 HYPERPARAMETERS

In this section, we present a comprehensive overview of the hyperparameters used in our experiments. It's important to note that the number of training samples was carefully selected to ensure that the amount of training data remained consistent across all methods.

**PPO**   Finetuning LLMs using PPO is known to be sensitive to hyperparameter selection, and finding the optimal settings is critical for achieving strong performance. To ensure the robustness of our study, we explored hyperparameter values reported in recent studies (Shao et al., 2024; Zheng et al., 2023; Ivison et al., 2024; Huang et al., 2024) and conducted various sweeps across a wide range of values to identify the best configuration for our tasks and models. The full set of hyperparameters, along with their respective search spaces, is detailed in Table 1.

**VinePPO**   We utilized the same hyperparameter setup as in the PPO implementation (Table 1) for VinePPO. As outlined in Section 5, the number of MC samples, $K$, was set to 9 for all experiments.

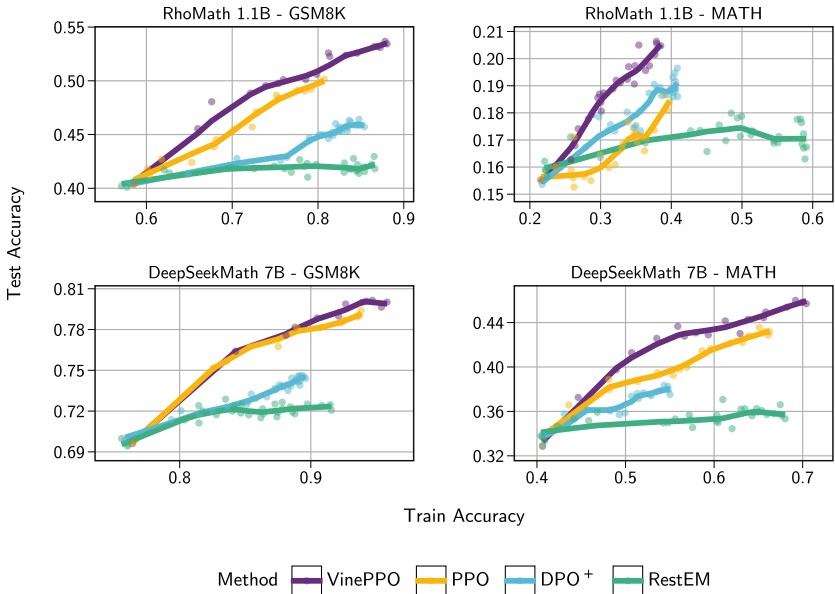

Figure C.4: A scatter plot showing the relationship between achieved training accuracy and test accuracy at various checkpoints throughout training. This plot highlights the dynamics of overfitting and generalization across different methods. As we progress from no credit assignment to accurate credit assignment—from RestEM to DPO$^+$, PPO, and finally VinePPO—generalization improves and overfitting decreases. In other words, by treating the training dataset as a resource, VinePPO achieves higher test accuracy per unit of training data consumed. Note that all these are fully trained. Note that the training accuracy does not reach 100 percent due to several factors, including mechanisms like the KL penalty in DPO$^+$, PPO, and VinePPO, the reset to the base model in RestEM, or the absence of any correct self-generated responses for certain questions.

**RestEM**   To ensure fair comparison we equalize the number of sampled responses for training between our RestEM run and our PPO runs. Therefore, in each RestEM iteration we sample 8 responses per prompt and train for 8 epochs on the correct responses. To enhance RestEM's performance, we also conducted a sweep of other reasonable parameters(Table 2). This included increasing the number of samples to expand the training data and reducing the number of correct responses per question to minimize overfitting.However, we observed no significant improvement .

**DPO$^+$ (DPO-Positive)**   We adopted the same hyperparameters as those used by Setlur et al. (2024). In addition, we conducted a search for the optimal value of $\beta$ to see if using the same $\beta$ as in our PPO experiments would yield better performance than the values they recommended. To maintain a fair comparison, we ensured that the number of training samples in our DPO$^+$ runs matched those in our PPO run where we trained for eight epochs, with each epoch consisting of training on eight responses per question. To match this, we generated 64 pairs of positive and negative responses given 64 self-generated responses from the base model. (Table 3)

## C.7   TRAIN VS. TEST DURING TRAINING

When training on reasoning datasets, the training data can be viewed as a finite resource of learning signals. Algorithms that exhaust this resource through memorization tend to generalize less effectively on the test set. As we move from RL-free methods or less accurate credit assignment towards more accurate credit assignment, or full reinforcement learning—from RestEM to DPO, PPO, and finally VinePPO—the model demonstrates higher test accuracy gains per unit of training data consumed. This trend is illustrated in Figure C.4.

Table 4: Average time spent per each training step for different methods and models measured for MATH dataset

| Method | Model | Hardware | Average Training Step Time (s) |
|--------|-------|----------|-------------------------------|
| PPO | RhoMath 1.1B | $4 \times$ Nvidia A100 80GB | 80 |
| VinePPO | RhoMath 1.1B | $4 \times$ Nvidia A100 80GB | 380 |
| PPO | DeepSeekMath 7B | $8 \times$ Nvidia H100 80GB | 312 |
| VinePPO | DeepSeekMath 7B | $8 \times$ Nvidia H100 80GB | 583 |

## C.8 COMPUTE

All experiments were conducted using multi-GPU training to efficiently handle the computational demands of large-scale models. For the RhoMath 1.1B model, we utilized a node with $4 \times$ Nvidia A100 80GB GPUs to train both PPO and VinePPO. For the larger DeepSeekMath 7B model, we employed a more powerful setup, using a node with $8 \times$ Nvidia H100 80GB GPUs. Additionally, for training DeepSeekMath 7B models with the RestEM approach, we used a node with $4 \times$ Nvidia A100 80GB GPUs. The average training step time for each method on the MATH dataset is presented in Table 4.

## C.9 SOFTWARE STACK

Both PPO and VinePPOrequire a robust and efficient implementation. For model implementation, we utilize the Huggingface library. Training is carried out using the DeepSpeed distributed training library, which offers efficient multi-GPU support. Specifically, we employ DeepSpeed ZeRO stage 0 (vanilla data parallelism) for RhoMath 1.1B and ZeRO stage 2 (shared optimizer states and gradients across GPUs) for DeepSeekMath 7B . For trajectory sampling during RL training, we rely on the vLLM library (Kwon et al., 2023), which provides optimized inference for LLMs. Additionally, VinePPOleverages vLLM to generate Monte Carlo samples for value estimation. This software stack ensures that our experiments are both efficient and reproducible. For instance, during VinePPO training, we achieve an inference speed of up to 30K tokens per second using $8 \times$ Nvidia H100 GPUs with the DeepSeekMath 7B model.

## C.10 REPRODUCIBILITY

In this study, all experiments were conducted using open-source libraries, publicly available datasets, and open-weight LLMs. To ensure full reproducibility, we will release both Singularity and Docker containers, equipped with all dependencies and libraries, enabling our experiments to be run on any machine equipped with NVIDIA GPUs, now or in the future. Additionally, we will make our codebase publicly available on GitHub at `https://www.omitted.link`.

# D FULL RESULTS

## D.1 TRAINING PLOTS

In this section, we present additional training plots for both PPO and VinePPO on the GSM8K dataset, as shown in Figure D.5. Figure D.6 further illustrates the trade-off between accuracy and KL divergence, while Figure D.7 highlights the computational efficiency of the models[7].

We observe consistent patterns with the results reported in Section 6. Although the performance gap for the DeepSeekMath 7B model is narrower on GSM8K, VinePPO still higher accuracy with significantly lower KL divergence and faster per-iteration time (this happens because responses to GSM8K problems are typically shorter, making MC estimation quite fast).

---

[7]For GSM8K runs of RhoMath 1.1B , different hardware was used, making direct comparison of wall-clock time not feasible.

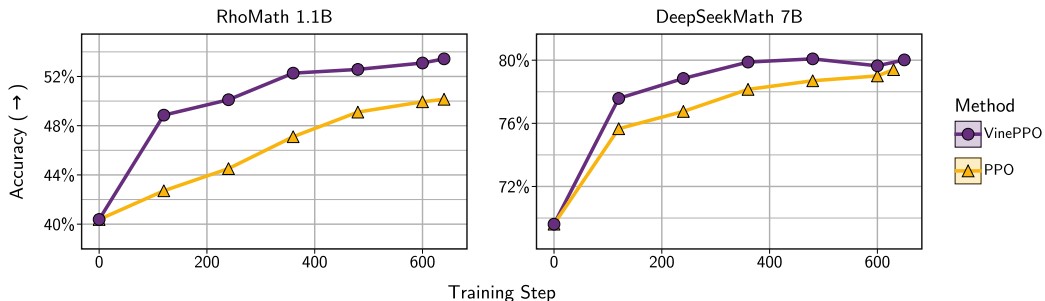

Figure D.5: Comparison of the training behavior between VinePPO and PPO. VinePPO demonstrates consistently higher accuracy throughout the training on the GSM8K dataset. Refer to Figure 5 for MATH dataset.

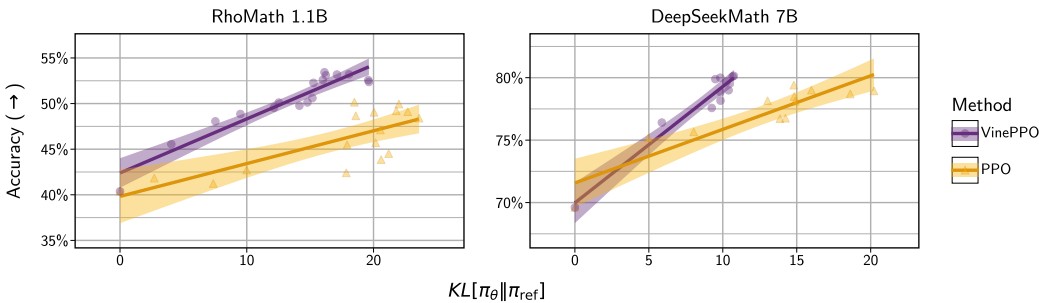

Figure D.6: Task accuracy as a function of KL divergence during training on the GSM8K dataset. VinePPO significantly higher accuracy per KL. Refer to Figure 6 for MATH dataset.

## D.2   VALUE PREDICTION ANALYSIS

In this section, we provide additional plots for value analysis. Specifically, Figures D.8 to D.11 demonstrates these plots for on the MATH dataset, and Figures D.12 to D.15 on the GSM8K dataset.

Furthermore, we present the prediction error per step in Figures D.16 to D.19.

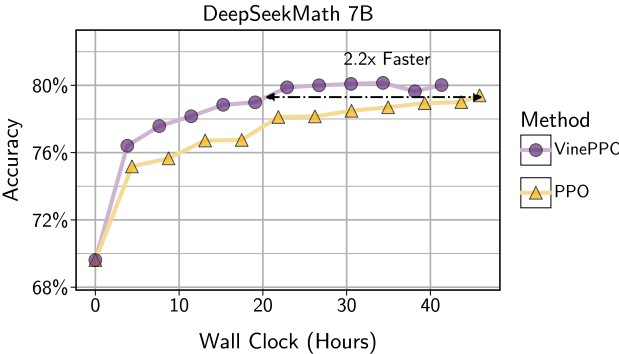

Figure D.7: Accuracy vs. Wall Clock Time for both methods measured on the same hardware throughout the entire training. Since the responses to GSM8K problems are short, VinePPO is even faster per-iteration in our setup and it reaches PPO's peak performance in fewer iterations and less overall time.

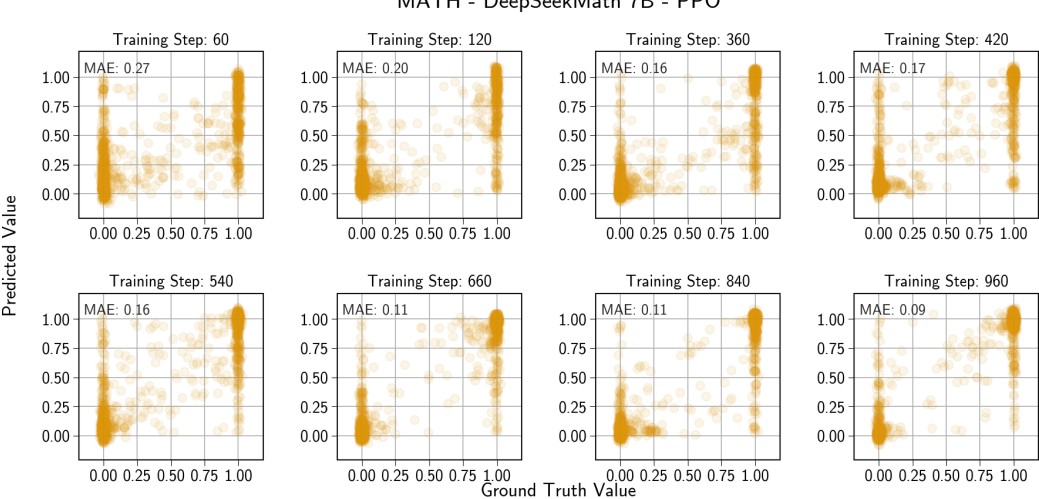

Figure D.8: Distribution of predicted values for each state vs. ground truth (computed using 256 MC samples) during training. MAE denotes the Mean Absolute Error (MAE).

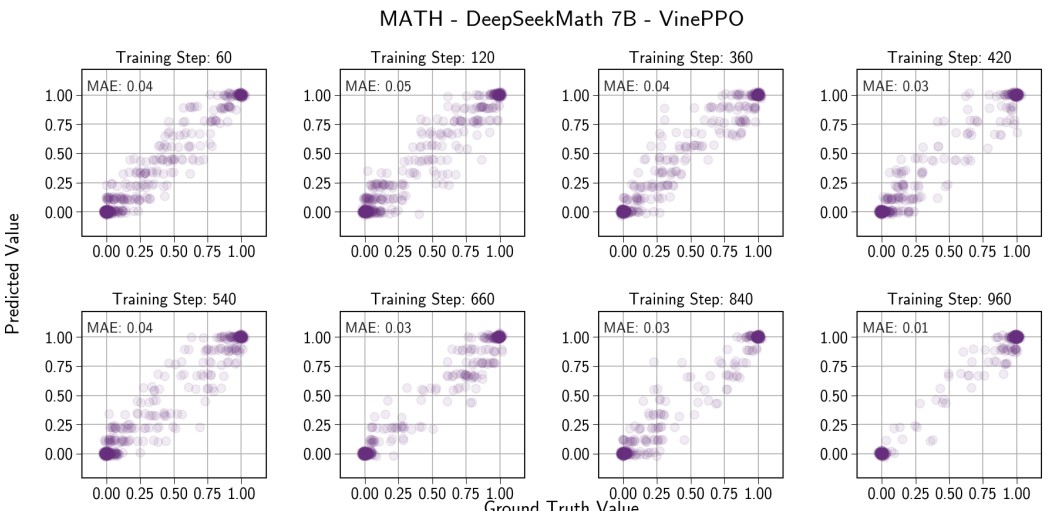

Figure D.9: Distribution of predicted values for each state vs. ground truth (computed using 256 MC samples) during training. MAE denotes the Mean Absolute Error (MAE).

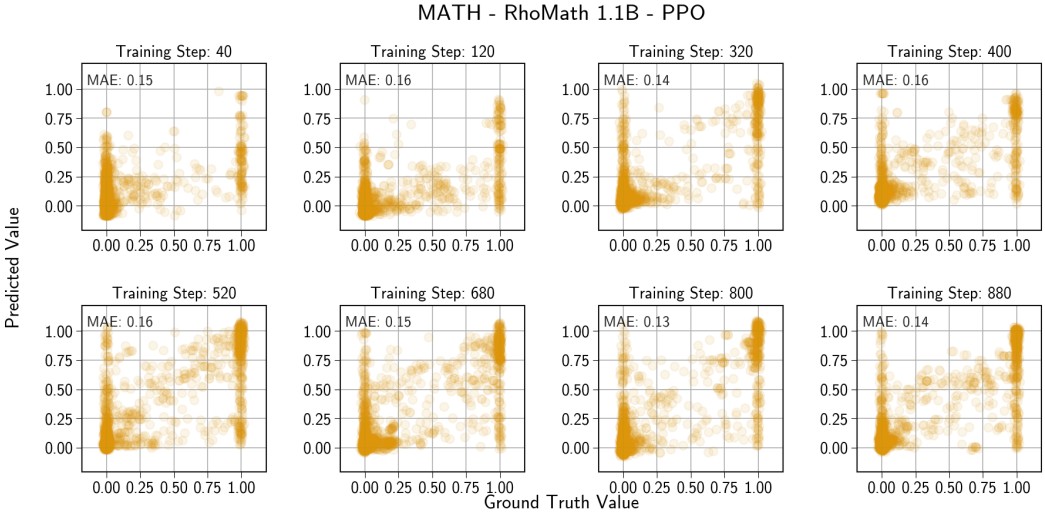

Figure D.10: Distribution of predicted values for each state vs. ground truth (computed using 256 MC samples) during training. MAE denotes the Mean Absolute Error (MAE).

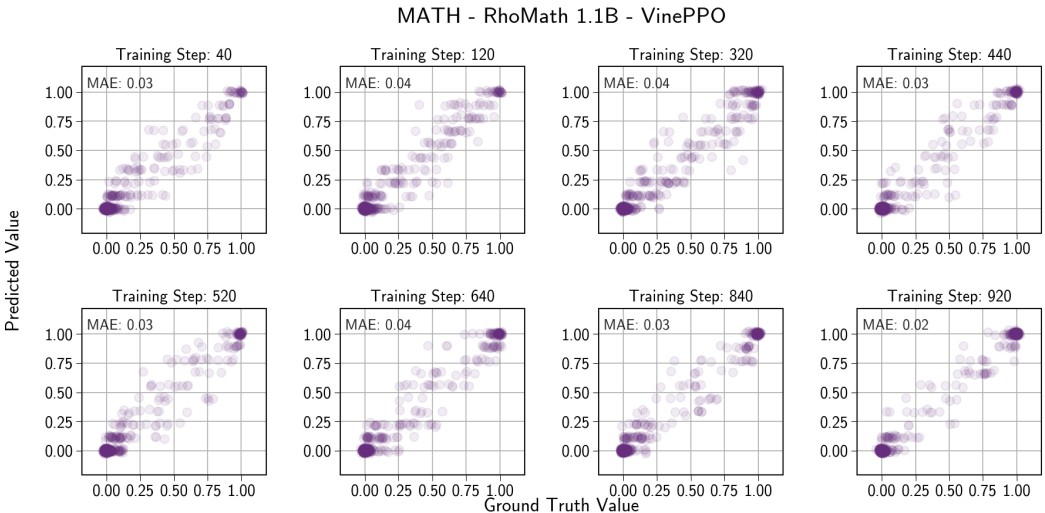

Figure D.11: Distribution of predicted values for each state vs. ground truth (computed using 256 MC samples) during training. MAE denotes the Mean Absolute Error (MAE).

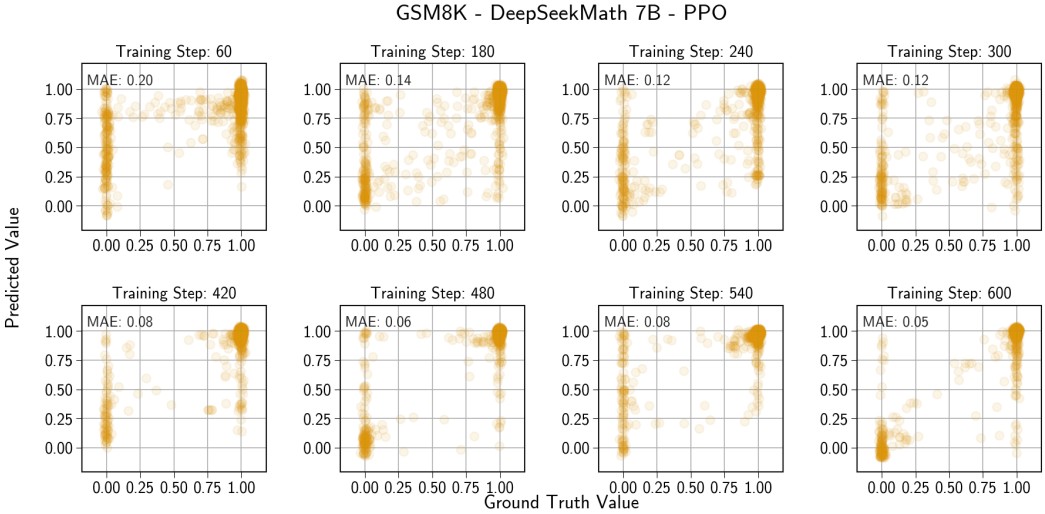

Figure D.12: Distribution of predicted values for each state vs. ground truth (computed using 256 MC samples) during training. MAE denotes the Mean Absolute Error (MAE).

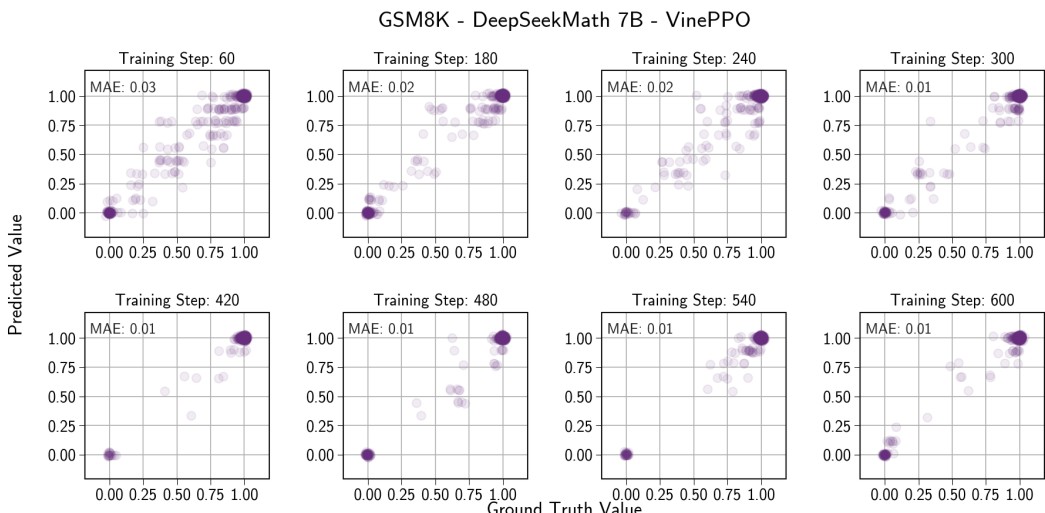

Figure D.13: Distribution of predicted values for each state vs. ground truth (computed using 256 MC samples) during training. MAE denotes the Mean Absolute Error (MAE).

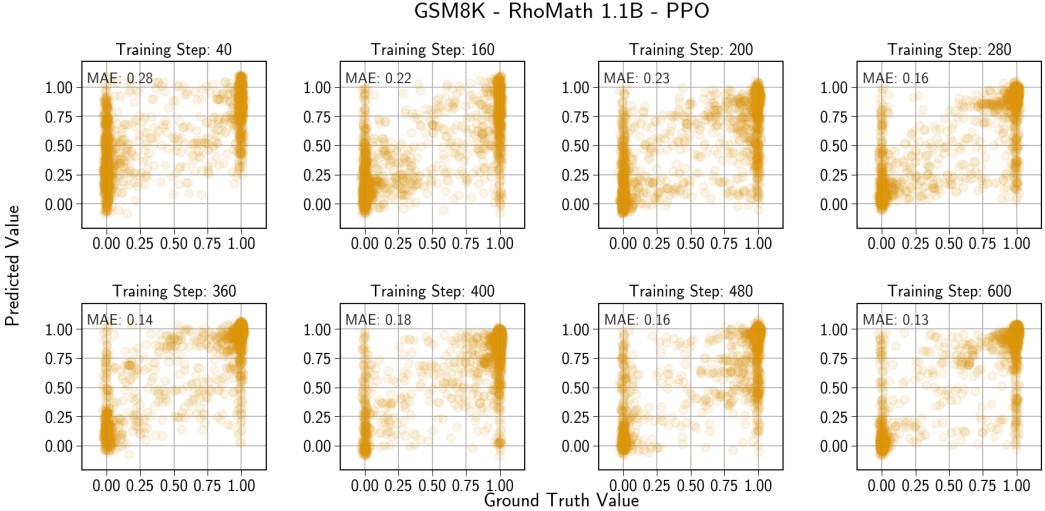

Figure D.14: Distribution of predicted values for each state vs. ground truth (computed using 256 MC samples) during training. MAE denotes the Mean Absolute Error (MAE).

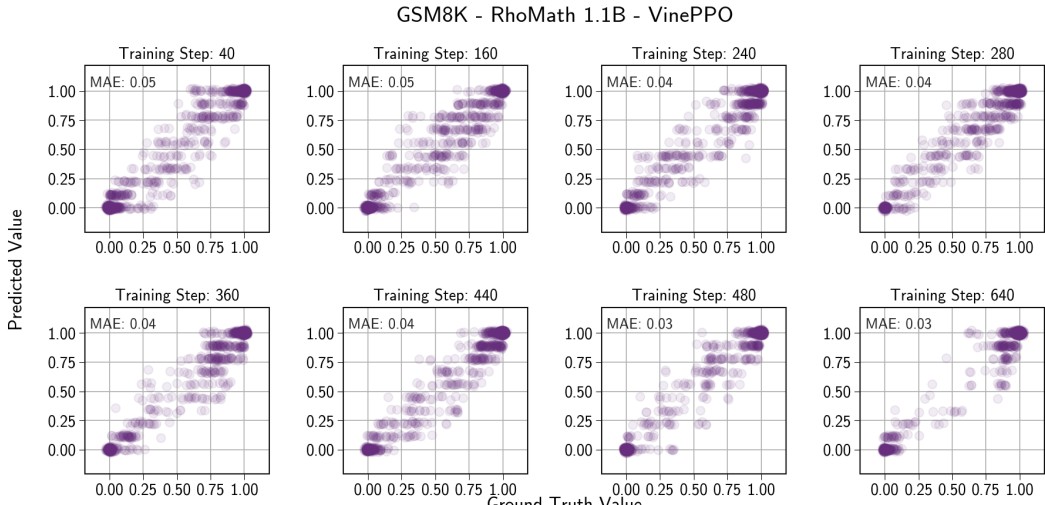

Figure D.15: Distribution of predicted values for each state vs. ground truth (computed using 256 MC samples) during training. MAE denotes the Mean Absolute Error (MAE).

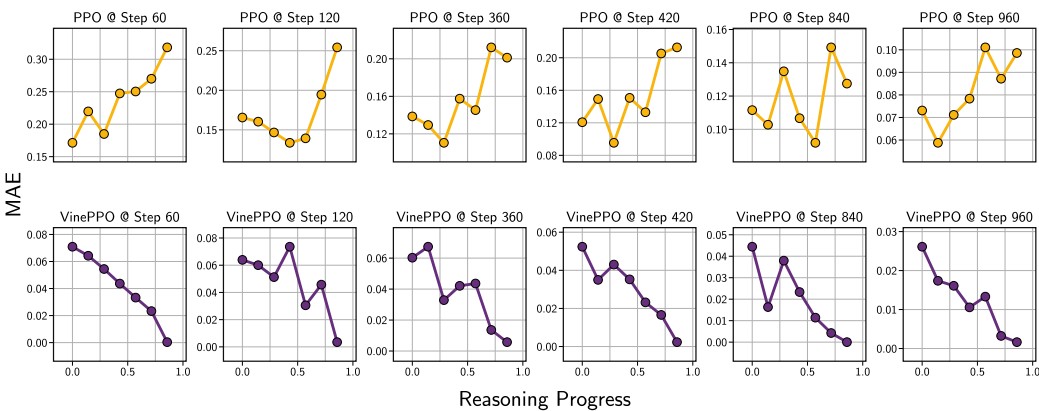

Figure D.16: Visualizing the Mean Absolute Error (MAE) of the value predictions in different point of reasoning chain, plotted for DeepSeekMath 7B on MATH dataset.

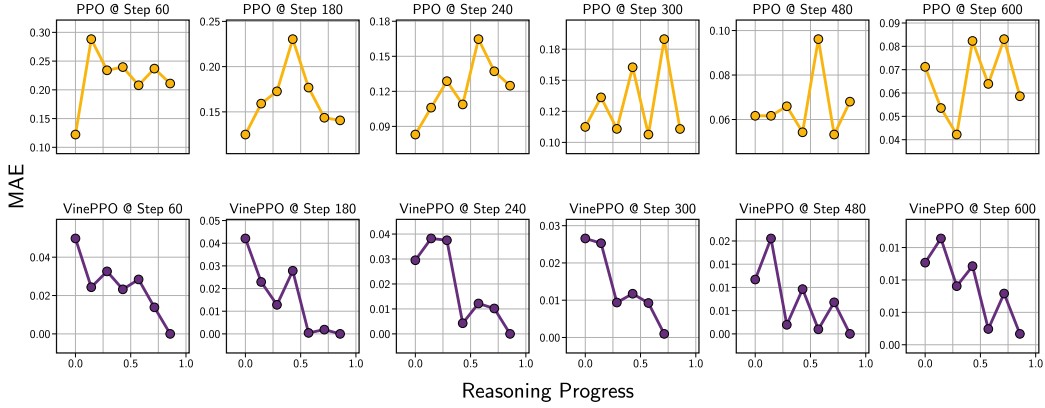

Figure D.17: Visualizing the Mean Absolute Error (MAE) of the value predictions in different point of reasoning chain, plotted for DeepSeekMath 7B on GSM8K dataset.

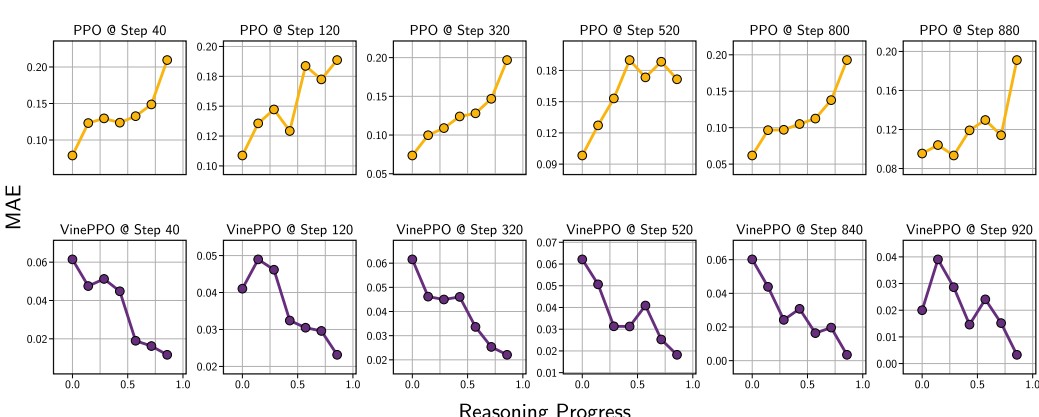

Figure D.18: Visualizing the Mean Absolute Error (MAE) of the value predictions in different point of reasoning chain, plotted for RhoMath 1.1B on MATH dataset.

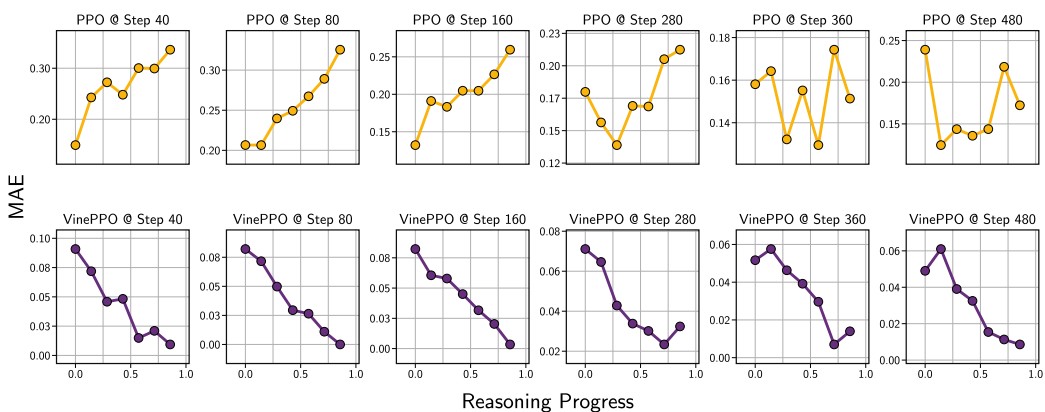

Figure D.19: Visualizing the Mean Absolute Error (MAE) of the value predictions in different point of reasoning chain, plotted for RhoMath 1.1B on GSM8K dataset.

# Post-Submission Updates

- RLOO and GRPO Baselines (Appendix E)
- Updated Compute Efficiency Plots (Appendix F)
    - RLOO and GRPO Efficiency (Appendix F)
    - Effect of $K$ in VinePPO's Efficiency (Appendix F)
- Updated Value Prediction Analysis (Appendix G)
    - Explained Variance and Mean Absolute Error (Appendix G)
- More Examples of Advantages in VinePPO (Appendix H)
- Difference Between Bias in Estimated Values and Bias in Policy Gradient (Appendix I)
- Updated "C.9 Software Stack" section (Appendix J)

# E    RLOO AND GRPO BASELINES

As requested by the reviewers, we included RLOO and GRPO as baselines and trained RhoMath 1.1B on GSM8K and MATH using these methods. As shown in Figure E.20, both RLOO and GRPO lag behind VinePPO. Comprehensive results and analysis are provided in Figures E.20 and E.21 and F.20.1

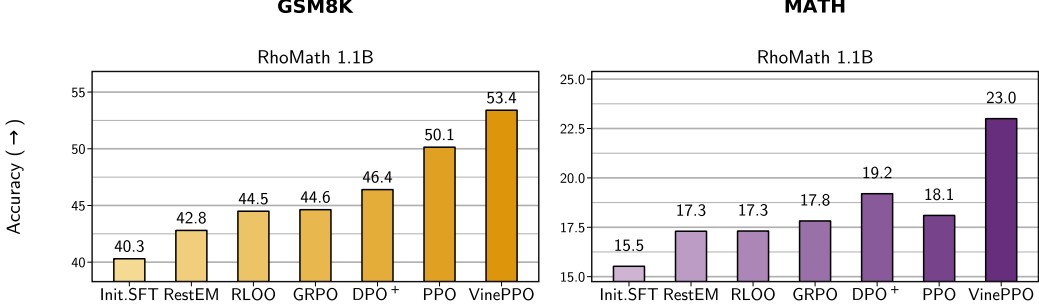

Figure E.20: **Pass@1 Performance of RLOO and GRPO Baselines** RLOO and GRPO outperform RestEM and match PPO on MATH but underperform PPO on GSM8K. VinePPO consistently surpasses all baselines. This is expected as RLOO and GRPO lack fine-grained credit assignment and use a shared baseline for all tokens. Their training is also less stable than VinePPO and PPO, requiring a higher KL coefficient. This instability likely stems from high bias in value estimates, leading to high-variance gradients. See analysis in Figures G.24 and G.25

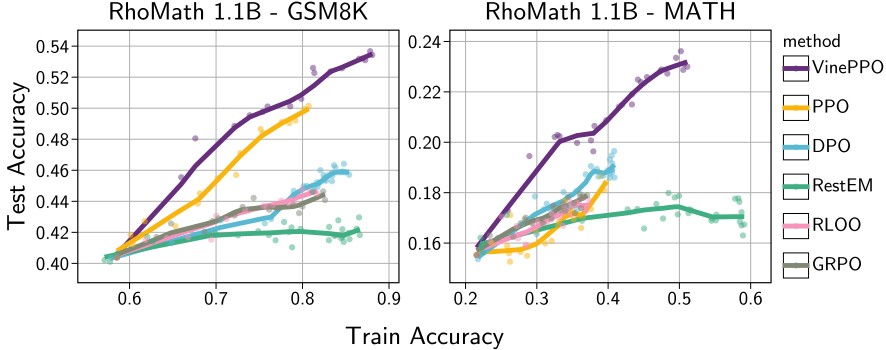

Figure E.21: **Train vs. Test Accuracy** This figure illustrates the generalization dynamics of various methods. VinePPO demonstrates superior generalization compared to all other baselines.

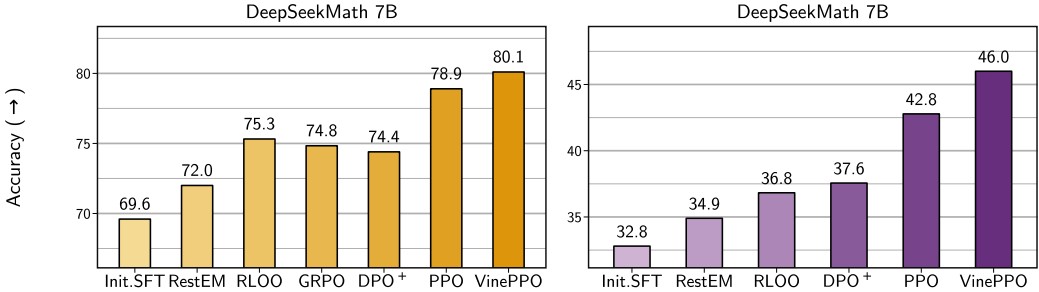

Figure E.20.1: **Pass@1 Performance of RLOO and GRPO Baseline on DeepSeekMath 7B** RLOO outperforms RestEM and DPO$^+$ but still underperforms both PPO and VinePPO on GSM8K (left). In MATH, which is a more challenging task, RLOO underperform both PPO and VinePPO (right).

# F UPDATED COMPUTE EFFICIENCY PLOTS

**RLOO and GRPO**  To evaluate the computational efficiency of these methods, we plotted test accuracy against wall-clock time during training in Figure F.22.

**Effect of $K$ in VinePPO**  In addition to analyzing final performance in Figure 4, we examine the impact of $K$ on computational efficiency in Figure F.23.

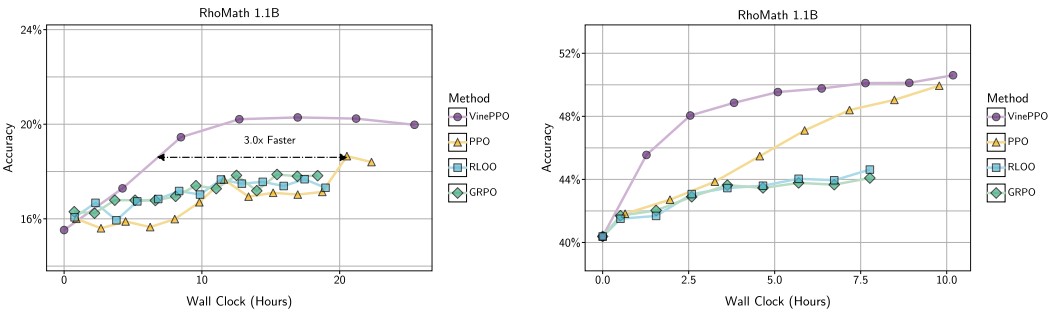

(a) performance on MATH  (b) performance on GSM8K

Figure F.22: **Compute Efficiency of RLOO and GRPO.** Accuracy vs. Wall Clock Time for all methods, measured on the same hardware. On MATH, VinePPO reaches the peak performance of RLOO and GRPO 2.7x and 2.2x faster, respectively, using identical computational resources. Notably, on GSM8K, even PPO—despite training an additional network—outperforms RLOO and GRPO in efficiency.

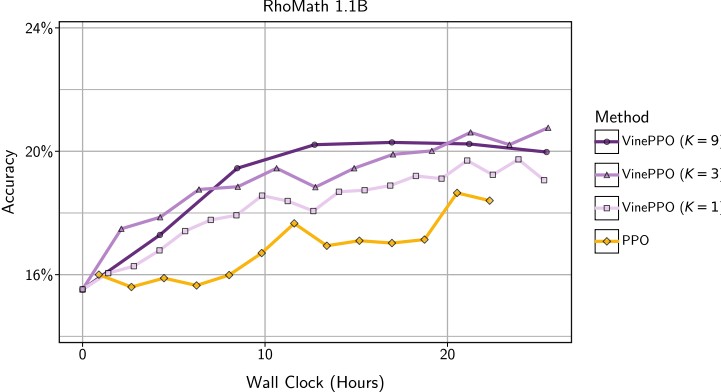

Figure F.23: **Effect of $K$ on compute efficiency of VinePPO.** Accuracy vs. Wall Clock Time for runs with different $K$ values, measured on the same hardware. Generally, VinePPO with higher $K$ achieves greater efficiency. VinePPO($K$=9) slightly outperforms VinePPO($K$=3), while both significantly surpass VinePPO($K$=1). Despite higher $K$ requiring nearly linear increases in computation, this result highlights the strong impact of low-variance value estimates on training, which shifts the trade-off toward improved efficiency with more samples.

# G UPDATED VALUE PREDICTION ANALYSIS

RLOO and GRPO use a shared baseline for all tokens in a response, resulting in high bias in value estimation for individual steps. To illustrate this, we follow the protocol in Section 7 and present the distribution of value predictions and the mean absolute error (MAE) across reasoning steps in Figures G.24 and G.25.

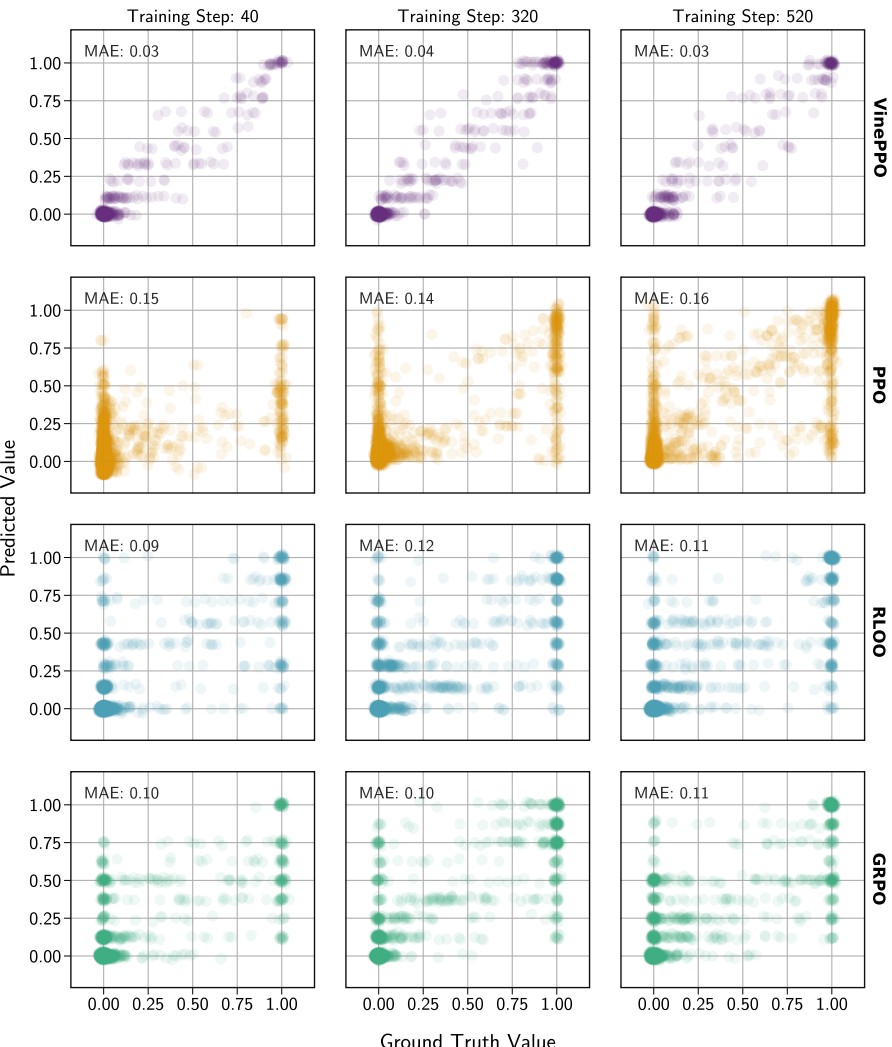

Figure G.24: **Value prediction analysis of VinePPO, PPO, RLOO, and GRPO during training**. Distribution of predicted values for each state vs. ground truth (computed using 256 MC samples) during training for RhoMath 1.1B on the MATH dataset, highlighting the nature of errors. While RLOO and GRPO exhibit slightly lower MAE compared to PPO, their errors are still significantly higher than VinePPO. Additionally, RLOO and GRPO estimates show a high bias, frequently assigning high values to states with a low probability of successfully completing the solution and vice versa. This is expected, as RLOO and GRPO inherently assign the same value or baseline to all steps in a response, lacking fine-grained credit assignment.

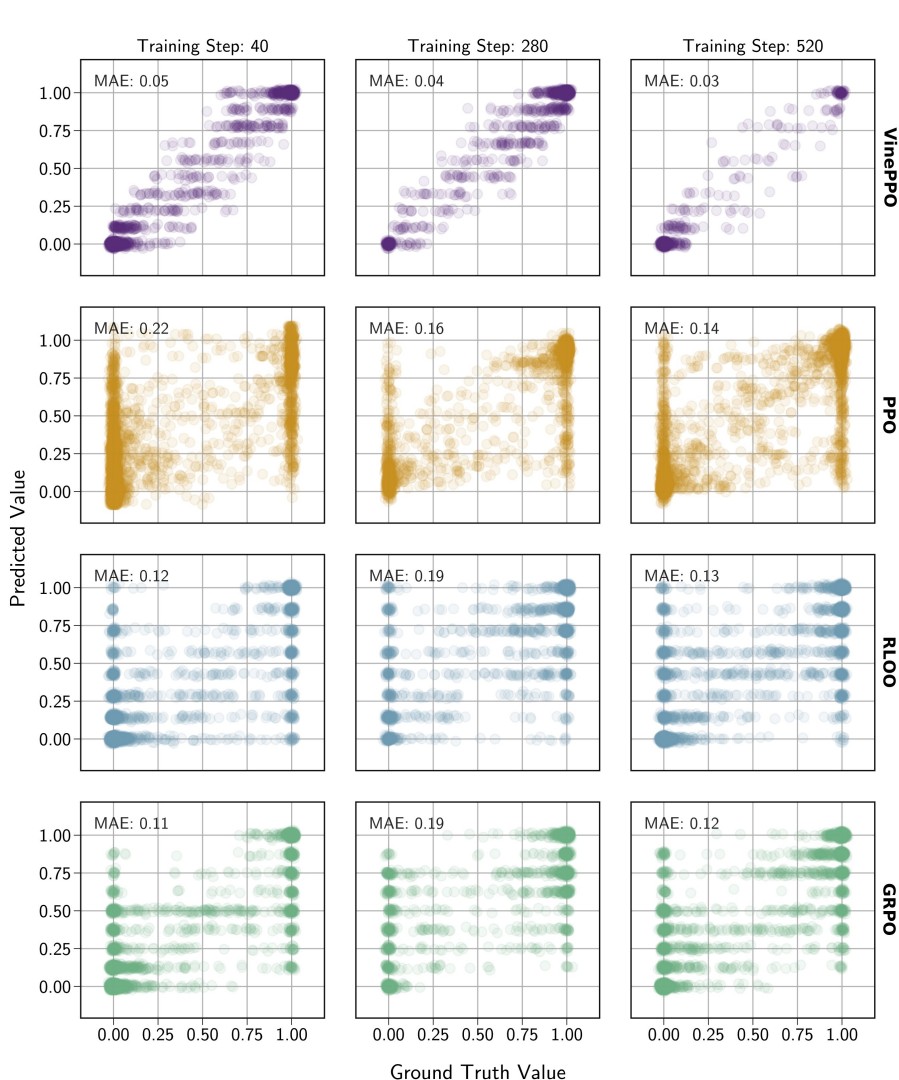

Figure G.25: Distribution of predicted values for each state vs. ground truth (computed using 256 MC samples) during training for RhoMath 1.1B on the GSM8K dataset. Similar to Figure G.24, RLOO and GRPO exhibit lower MAE than PPO but significantly higher than VinePPO.

## G.1 EXPLAINED VARIANCE AND MEAN ABSOLUTE ERROR (MAE)

In addition to the analysis in Appendix G, we quantify the accuracy of value predictions using explained variance and mean absolute error during training, as shown in Figure G.26.

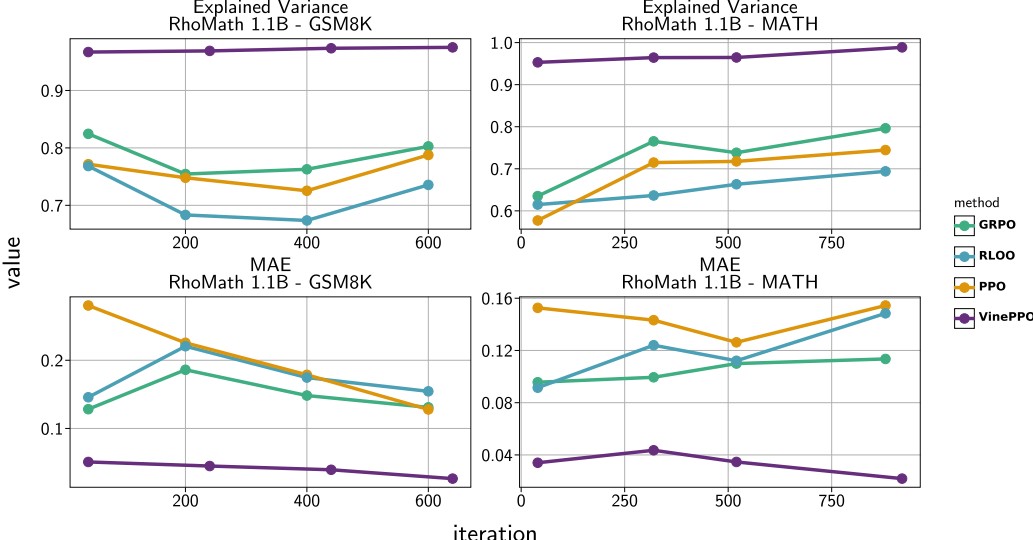

Figure G.26: **Explained Variance and Mean Absolute Error of values**. VinePPO demonstrates higher explained variance in value predictions and lower mean absolute error compared to RLOO, GRPO, and PPO across both datasets. Additionally, PPO's value predictions show non-negative explained variance values close to one, indicating stable and effective training. Furthermore, the value network in PPO achieves lower error on GSM8K at the end of training than RLOO and GRPO, highlighting that PPO's value network can outperform simplistic baselines in certain scenarios.

# H    MORE EXAMPLES OF ADVANTAGES IN VINEPPO

In addition to Figure 1, we provide more examples of effective advantage computation of Vine PPO in Figures H.27 to H.29.

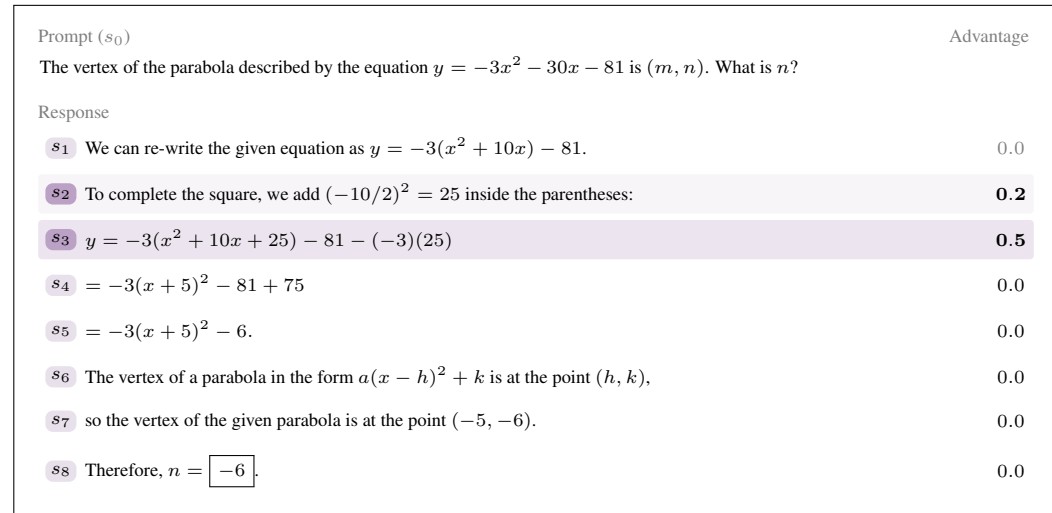

Example of Advantages Computed by VinePPO

Figure H.27: Steps $s_2$ and $s_3$ provide a critical insight for advancing the solution since they provide a way to organize the equation.

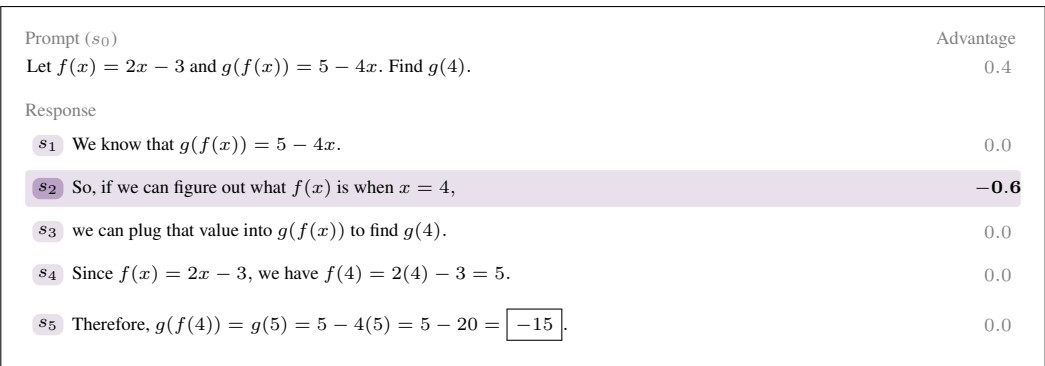

Example ofAdvantages Computed by VinePPO

Figure H.28: This example demonstrates an error in step $s_2$ where the correct step would be ...*what $x$ is when $f(x) = 4$.* This mistake prevents to model to successfully solve the question after $s_2$.

| Prompt ($s_0$) | Advantage |
|---|---|
| Write the expression $\frac{4+3c}{7} + 2$ as a single fraction. | |
| Response | |
| $s_1$   We get a common denominator of 7: | 0.0 |
| $s_2$   $\frac{4+3c}{7} + 2 = \frac{4+3c}{7} + \frac{2 \cdot 7}{7}$ | 0.0 |
| $s_3$   $= \frac{4+3c}{7} + \frac{14}{7}$ | $-0.3$ |
| $s_4$   $= \frac{4+3c+14}{7}$ | 0.4 |
| $s_5$   $= \boxed{\dfrac{18+3c}{7}}$. | 0.0 |

Example of Advantages Computed by VinePPO

Figure H.29: Steps $s_3$ and $s_4$, despite impacting the probability of solving the question, they are not meaningfully incorrect or insightful.

# I  DIFFERENCE BETWEEN BIAS IN VALUE ESTIMATIONS AND IN POLICY GRADIENT

Note that when $\lambda = 1$, the value estimates are used solely as a baseline. It is well-known that, in this case, the policy gradient Eq 2 provides an unbiased estimate of the true values. However, it is important to emphasize that the value estimates themselves can still be biased. Consequently, the fact that the policy gradient is unbiased does not guarantee that the value estimates used to compute the advantages are unbiased estimators of the true value of a given state.

# J  UPDATED "C.9 SOFTWARE STACK" SECTION

Both PPO and VinePPOrequire a robust and efficient implementation. For model implementation, we utilize the Huggingface library. Training is carried out using the DeepSpeed distributed training library, which offers efficient multi-GPU support. Specifically, we employ DeepSpeed ZeRO stage 0 (vanilla data parallelism) for RhoMath 1.1B and ZeRO stage 2 (shared optimizer states and gradients across GPUs) for DeepSeekMath 7B . For trajectory sampling during RL training, we rely on the vLLM library (Kwon et al., 2023), which provides optimized inference for LLMs. Additionally, VinePPOleverages vLLM to generate Monte Carlo samples for value estimation. Specifically, after each RL training iteration, the current policy's checkpoint is loaded into vLLM. Then, we use vLLM's serving API to sample new trajectories and also Monte Carlo Samples for VinePPO's value estimation. In our setup, we spawn a separate vLLM engine on each GPU rank. This would allow for data parallelism during both sample generation and training. This software stack ensures that our experiments are both efficient and reproducible. For instance, during VinePPO training, we achieve an inference speed of up to 30K tokens per second using $8 \times$ Nvidia H100 GPUs with the DeepSeekMath 7B model.

