# OpenReview forum: "VinePPO: Unlocking RL Potential For LLM Reasoning Through Refined Credit Assignment"
_ICLR.cc/2025/Conference — Submitted to ICLR 2025_

### Official Review · Reviewer_3mkL · 2024-11-03

**Soundness:** 2
**Presentation:** 3
**Contribution:** 2
**Rating:** 3
**Confidence:** 3

**Summary:**

The key motivation of this manuscript is to locate and solve the problem, while PPO is finetuning LLM，the value network is inaccurate and has high variances. It finds that in heavy and complex reasoning tasks, PPO barely outperform a random baseline due to this issue. Thus, this paper proposes a simple and straightforward approach, so called VinePPO, which computes the value using unbiased Monte Carlo estimation and improve the credit assignment. Many experiments on MATH and GSM8K datasets with RhoMath 1.1B and DeepSeekMath 7B, show that the proposed VinePPO can consistently outperforms PPO and other RL-free baselines.

**Strengths:**

1.	The authors found the problem via systematical evaluation that, inaccurate value estimation can limit PPO’s ability to finetune LLMs in complex reasoning tasks. It can’t reflect the real reward and importance. This problem results in the barely fair performance compared with a random baseline.
2.	This paper proposes the VinePPO via utilizing MC samples to compute value in the PPO pipeline, and the value of a state can be estimated by the average return of K sampled trajectories from the state.
3.	The experiments and analytics are convincing. The results of VinePPO is better than PPO, via improved credit assignment.

**Weaknesses:**

1.	The proposed VinePPO is a straightforward method to use MC estimation. However, MC has been studied for a long time. It has zero bias, but also has high variance and computational efficiency problem.
2.	This paper adopts the math reasoning problem. The state is the concatenation of input prompt and generated tokens, so the following trajectories can be sampled from any state s, and then MC computation can work. But, if the problem is more complex, not simple math problem, MC might not work, because long trajectory or low efficiency.
3.	In VinePPO, K is very important, because accurate MC estimation needs K be large enough, which also would cause low efficient issue.
4.	Based on the discussion above, the generalizability of VinePPO is not analysed and  solved in the paper.

**Questions:**

1.	The influence of K needs to be discussed, from both performance and efficiency.
2.	As MC is a method with high variance, does VinePPO outperform PPO when MC estimation is not inaccurate?
3.	In Fig 9, the ground truth is chosen as results via 256 MC samples. Is this reasonable?
4.	It might be more convincing to provide the resulting of credit assignment. For example, are critical steps detected by VinePPO?
5.	Some equations are not clear, for example:$S_{t+1} =  s_t;[a_t]$

**Details Of Ethics Concerns:**

None.

---

> ### Author Response · Authors · 2024-11-19
> **Rebuttal**
>
> We thank the reviewer for their time and feedback.
>
> We should note that we don’t completely agree with the provided summary. In our work we don’t show *“PPO barely outperforms a random baseline”*. Instead, we believe the reviewer meant that *“The value network in PPO performs just slightly better than a random choice when trying to rank the values of actions.”*
>
> We now address the concerns raised in the review.
>
> **Variance and efficiency of MC estimation**
>
> While variance and computation efficiency are valid concerns for any MC-based estimation, we studied these extensively in Sec. 6.4 of the initial submission (noted by review 1twu): specifically, in Figs. 4, 7, D.7 of the original paper, and now in Fig. F.22 and F.23 of the updated draft. Across all experiments, VinePPO with its MC value estimation consistently outperformed all baselines (including new ones suggested during the Rebuttal) in both efficiency and accuracy, as noted by reviewer wh2b.
>
> Also, while MC has been studied for a long time, we note that, as reviewer S7we highlighted, “The idea of Monte Carlo estimates, although it has been used in traditional RL tasks, is novel for PPO in the context of LLMs.” Moreover, MC value estimation in policy-gradient method is limited to environments that allow intermediate state resets, rare in classic RL environments [1]
>
> **Importance of K**
>
> The impact of K is studied in Fig. 4 of the original draft. Additionally, we study the effect of K in efficiency in Fig. F.23 of the updated paper. As noted by reviewer s7we, even small values of K (e.g. 1 or 3) perform well and outperform PPO with its value network (Fig. 4). Additionally, as shown in Fig. 23, higher values of K actually improve the wall-clock time efficiency in terms of reaching a target accuracy.
>
> **Generalizability**
>
> VinePPO is built on standard PPO from RLHF and introduces no additional assumptions to the RL formulation. Our empirical results demonstrate that VinePPO is more efficient than PPO, making it broadly applicable to RL problems in language environments where PPO is typically used, as noted by reviewer S7WE.
>
> We highlight that the MATH dataset, our primary evaluation suite, includes competition-level problems with long trajectories (up to 2500 toks). Common RLHF methods like DPO often struggle in such tasks [2], demonstrating the challenging nature of our setup.
>
> **Questions**
> > Q1. influence of K on  performance and efficiency
>
> As mentioned earlier, the impact of K on performance is analyzed in Fig. 4 of the original paper, showing that increasing K improves test accuracy. The effect of K on wall-clock efficiency is detailed in Fig. F.23, where VinePPO with higher K values shows greater efficiency. Specifically, VinePPO with K=9 is slightly more efficient than K=3, and both significantly outperform K=1 in reaching a target accuracy.
> > Q2. does VinePPO outperform PPO when MC estimation is not inaccurate?
>
> If very few MC estimates are used, high variance could theoretically hinder training. However, we did not observe this empirically. As shown in Fig. 4 and F.23, even with one MC sample, VinePPO outperforms PPO in both final performance and wall-clock efficiency (as noted by reviewer S7We)
> > Q3. In Fig 9, the ground truth is chosen as results via 256 MC samples. Is this reasonable?
>
> Yes. In our tasks, the value of a state follows a Bernoulli distribution representing the probability of successfully completing a solution. Assuming the maximum possible variance of this distribution (0.25), the variance of the ground truth estimator is 0.25 / 256 = 0.00097656, which is notably small.
> > Q4. are critical steps detected by VinePPO?
>
> Thank you for the suggestion. Beyond Fig. 1.a in the original paper, additional examples are in Figures H.27–H.29 of the updated draft. Fig. H.27 shows an insightful reasoning step with positive advantages recovered by VinePPO, while Fig. H.28 illustrates an erroneous step with negative advantages. However, high or low values reflect the policy’s likelihood of solving the problem, which may not always align with human judgments of correctness (see Fig. H.29).
> > Q5. Some equations are not clear, for example: St+1=st;[at]
>
> As noted in the background (lines 198–200), $s_t;[a_t]$ refers to “appending action $a_t$ to state $s_t$.” We will clarify this in the camera-ready version. If there are other equations requiring clarification, we are happy to address them too.
>
> Thank you for your effort again. Efficiency was a key focus of our work, and we conducted a thorough analysis of VinePPO’s efficiency in the paper. Across all experiments VinePPO consistently demonstrated superior efficiency and final performance compared to baselines. We hope this clarification and the additional empirical evidence address your concerns and encourage a fresh evaluation of our work.
>
> - [1] Trust Region Policy Optimization by Schulman et al. 2015
> - [2] SimPO: Simple Preference Optimization with a Reference-Free Reward by Meng et al. 2024

---

> > ### Comment · Reviewer_3mkL · 2024-11-26
> >
> > Thank you for the authors' responses and changes to my concerns, especially about the Variance  and Importance of K. I am glad to raise my score to 5.

---

> > > ### Author Response · Authors · 2024-12-02
> > > **Last Date of Reviewers Response**
> > >
> > > Dear Reviewer,
> > >
> > > Thank you again for considering our rebuttal. As today is the final day, we kindly ask if you could update your score in OpenReview using the "Edit" option in your official review to reflect your increased score.

---

> ### Author Response · Authors · 2024-11-21
> **Updates and Feedback Request**
>
> Dear Reviewer,
>
> Thank you for your time and feedback. With the end of the discussion phase approaching, we’re open to addressing any additional concerns or suggestions that could help us obtain a stronger evaluation. Especially, we updated the paper with additional efficiency plots on effect of K. Thank you once again for your time and effort in reviewing our work.

---

> ### Author Response · Authors · 2024-11-26
> **Feedback Request**
>
> Dear Reviewer,
>
> You raised efficiency as a concern regarding VinePPO. This was a primary focus of our work, and we extensively addressed it in the initial draft (see Section 6.4). During the rebuttal, we provided additional efficiency analysis as per your request. **All empirical results consistently demonstrate VinePPO’s superior efficiency, achieving higher test accuracy in less wall-clock time in every experiment (see Figures 7, F.22, and F.23)**. As today is the final day for updates, we would appreciate knowing if any concerns remain that influenced your evaluation.

---

> ### Author Response · Authors · 2024-11-27
> **Reflecting increased score in OpenReview**
>
> Dear Reviewer,
>
> Thank you for your attention to our rebuttal and for considering increasing your score. Could we kindly ask you to update your rating in OpenReview using the "Edit" button on your first comment (your official review), as verbal mentions don't reflect in the review system?

---

> ### Author Response · Authors · 2024-12-01
> **One day to Reviewer response deadline**
>
> Dear Reviewer,
>
> Thank you for your consideration of our rebuttal. As the reviewer response deadline is tomorrow, **we'd like to kindly remind you that your increased score is still not reflected in OpenReview**. We kindly request that you update your score using the "Edit" option in your official review since verbal mentions are not reflected in the system.
>
> We appreciate your time and effort!

---

### Official Review · Reviewer_1TWU · 2024-11-05

**Soundness:** 3
**Presentation:** 3
**Contribution:** 3
**Rating:** 6
**Confidence:** 4

**Summary:**

Large Language Models (LLMs) are increasingly being applied to complex reasoning tasks, ranging from solving math problems to developing code. The most common method for training these models is Proximal Policy Optimization (PPO), which addresses the credit assignment problem using a value network. However, this value network can be significantly biased, which may impact performance. The authors propose VinePPO, inspired by VineTRPO, to learn a value function from Monte Carlo (MC) samples. They demonstrate that the value function from PPO performs poorly, while the MC sample estimates of the value function show strong performance and leverage compute-efficient inference techniques.

**Strengths:**

- The author's observation that the issue with PPO was the value function estimates is very insightful, given that there has been a lot of work to replace PPO with new techniques.
- The paper was well written.
- The paper's experimental results provide a lot of interesting insights regarding issues around PPO's value function.
- The authors performed experiments across several tasks, model sizes, and model types.
- The authors' ablations studies show interesting pitfalls of the value function from PPO. Additionally, the authors clarify the tradeoff between VinePPO and PPO.

**Weaknesses:**

- I understand that the paper focuses on addressing the pitfalls of PPO; however, comparing it with RLOO [1] would provide practitioners with valuable context on which algorithm they might want to use in practice.
- The paper lacks details on how the inference engines were utilized to accelerate data gathering.

[1] Back to Basics: Revisiting REINFORCE Style Optimization for Learning from Human Feedback in LLMs by Ahmadian et al. 2024

**Questions:**

- Missing citations
   - RL + LLM: [1, 4]
   - RL: [2, 3, 5]
- How does VinePPO compare to the RLOO baseline as the value of K increases in RLOO?
-Did you do large-batch PPO updates? (Refer to [6] for the large-batch updates.) If you didn’t use the large-batch setting, essentially what you do is compute all the data statistics offline. This approach allows you to avoid loading the reward model onto the GPU, enabling you to increase your batch size much higher than if the reward model were loaded onto the GPU.
- Why is PPO more deterministic in early steps, while VinePPO is more deterministic in later steps, as mentioned in the "Error per reasoning step" section?
- Could you share a plot showing the "explained variance" of the value function you learn with normal PPO? (see [7])

[1] Learning to Generate Better Than Your LLM by Chang et. al 2023
[2] Exploring restart distributions by Tavakoli et al. 2018
[3] Data-efficient deep reinforcement learning for dexterous manipulation by Popov et al. 2017
[4] Dataset Reset Policy Optimization for RLHF by Chang et al 2024
[5] Mastering the game of Go with deep neural networks and tree search by Huang 2016
[6] SimPO: Simple Preference Optimization with a Reference-Free Reward by Meng et al. 2024
[7] http://joschu.net/docs/nuts-and-bolts.pdf

---

> ### Author Response · Authors · 2024-11-19
> **Rebuttal**
>
> We would like to thank the reviewer for their positive and insightful feedback. We will address mentioned points in detail as follows:
>
> **RLOO and GRPO Baselines**
>
> Thank you for the suggestion. As it was a common recommendation from reviewers, we updated the paper to include the result along with an in-depth analysis. We post this as a general comment. Please see *“Summary Response + RLOO and GRPO Baselines”* on OpenReview. A brief summary is provided below:
>
> RLOO and GRPO show a clear disadvantage compared to VinePPO. On GSM8K, they score 44.5% and 44.6%, respectively, while PPO achieves 50.1%, and VinePPO achieves 53.4%. A similar pattern is observed on MATH, where RLOO and GRPO score 17.3% and 17.8%, compared to PPO’s 18.1% and VinePPO’s 23.0%. Additionally, training RLOO and GRPO proved to be less stable; for instance, we found it necessary to use higher KL coefficients during hyperparameter tuning to prevent instability. Notably, when controlling for wall-clock time efficiency, we found VinePPO achieves their peak performance up to 2.7x faster.
>
>
> We also invite the reviewer to see the additional analysis included in the general comment, as we believe they offer interesting insights into the inner workings of RLOO and GRPO.
>
> **details on how the inference engines were utilized**
>
> We appreciate the reviewer’s interest in the technical details. We used the vLLM library [1] for fast inference and found that fortunately no special techniques were needed to achieve high throughput. Even 7B models can be deployed on each rank without any issues. That is, at every iteration current policy is loaded to the vLLM server and then sampling is done through vLLM serving API.  We will update Section C.9 (“Software Stack”) in the camera-ready version to include more details, Additionally, we plan to release our code with the camera-ready version, allowing others to use our generation pipeline.
>
> **Questions:**
> > **Q1.** Missing citations
>
> We thank the reviewer for their suggestion and we make sure to cite these works in the camera ready paper.
>
> > **Q2.** How does VinePPO compare to the RLOO baseline as the value of K increases in RLOO?
>
> Great question! Currently, we have focused our computational resources on running the RLOO and GRPO experiments with the 7B model. If time permits, we will conduct experiments with varying K values and share the results.
>
> > **Q3.** Did you do large-batch PPO updates?
>
> We thank the reviewer for their detailed suggestion. We are using a relatively large rollout batch size of 512 and mini-batch size of 64. Note that we do not have a reward model running on the GPU. In our tasks, the reward function is a Python program that compares the model's output with the ground truth.
>
> > **Q4.** Why is PPO more deterministic in early steps, while VinePPO is more deterministic in later steps, as mentioned in the "Error per reasoning step" section?
>
> We hypothesize that the value network relies on memorization. In the early steps, responses align closely with the training data, enabling accurate estimates. However, as responses progress, the number of possible sequences grows combinatorially, quickly moving out of the training data distribution. In contrast, MC estimation is not tied to the training data. At later steps, the LLM, conditioned on a long solution, becomes deterministic, requiring fewer MC samples for accurate estimation.
>
> > **Q5.** Could you share a plot showing the "explained variance" of the value function you learn with normal PPO?
>
> Thank you for suggesting the "explained variance" metric—we found it very insightful and computed this for other methods too. As shown in Figure G.26 of the updated paper, PPO’s value network predictions exhibit non-negative explained variance values close to one, reflecting healthy and effective training. Notably, VinePPO achieves higher explained variance in value predictions compared to PPO, RLOO, and GRPO.
>
>
> Thank you again for your valuable feedback; your suggestions have improved our paper. We hope we have adequately addressed all of your concerns and would be happy to clarify further if needed. With this in mind, we hope the reviewer increases their rating of our paper.
>
> - [1] Kwon, Woosuk, et al. "Efficient memory management for large language model serving with pagedattention." Proceedings of the 29th Symposium on Operating Systems Principles. 2023.

---

> ### Author Response · Authors · 2024-11-21
> **Updates and Feedback Request**
>
> Dear Reviewer,
>
> Thank you for your constructive comments. As the discussion phase is ending, we would love to address any remaining questions or incorporate further suggestions to obtain a better evaluation. Especially, we had already added RLOO and GRPO results for the 1B model and have now included a 7B result as it became available. We deeply appreciate your time and commitment to the review process.

---

> > ### Comment · Reviewer_1TWU · 2024-11-24
> >
> > Thank you for the responses to my questions and concerns, especially for plotting the "explained variance" metric for all of the algorithms discussed in the paper. I have no further questions and will keep my score the same.

---

> > > ### Author Response · Authors · 2024-11-27
> > >
> > > Dear Reviewer,
> > >
> > > Thank you for highlighting the relevant papers. They are indeed great papers. However, as today is the final day for updates, we kindly request your guidance in integrating them into our work. While we have added citations for [1] and [5], we would greatly value your input on the main angle of relevance and the most suitable placement for citing [2], [3], and [4].
> > >
> > > We also appreciate your valuable discussion about baselines. All of our RLOO experiments on all models (see Figure E.20.1 for 7B model and Figure E.20 for 1B model) and datasets are concluded now and in the updated paper showcasing VinePPO’s superior performance. We were wondering if our rebuttal has addressed your primary concern which was the RLOO baseline.
> > >
> > > Moreover, as this is the last day to update the paper, we wanted to know if there is any outstanding concern that prevents a stronger evaluation of our work?

---

> ### Author Response · Authors · 2024-12-01
> **One Day Left for Reviewer Response**
>
> Dear Reviewer,
>
> Thank you for your valuable insights and suggestions. As tomorrow is the final day for reviewer response, we wonder if you've had the chance to go over our previous response, and if you have any feedback on **the integration of highlighted papers**, and **any remaining concerns about the RLOO baseline**, or **other aspects that prevents a stronger evaluation of our work.**

---

### Official Review · Reviewer_S7We · 2024-11-08

**Soundness:** 3
**Presentation:** 4
**Contribution:** 2
**Rating:** 6
**Confidence:** 3

**Summary:**

The paper proposes vine-PPO, which uses Monte Carlo-based estimates to replace the value function. This approach is far more accurate and therefore performs better than the parameterized value function. Although the cost could be a concern, the authors argue that inference or generation is much faster due to many inference-optimized modules. Additionally, because of the rapid increase in performance, it may even be more efficient.

**Strengths:**

- The idea of Monte Carlo estimates, although it has been used in traditional RL tasks, is novel for PPO in the context of LLMs. I find it quite interesting that it can achieve superior results even with K=1.
- The applicability stemming from the fact that it only replaces the value function, allowing it to be used in many PPO-like methods, is highly beneficial.
- The analysis of the value function helps clarify the motivation.
- The proposed method is simple and easy to follow, and the paper is well-written.

**Weaknesses:**

- Fundamentally, I think the difference between your approach and GRPO [1] and RLOO [2] is that you have fine-grained value estimations by generating multiple responses from each intermediate group state. However, since this involves more computation, I wonder about the trade-offs compared to GRPO.
- This question arises because you do not compare your method with GRPO and RLOO. As these methods also employ similar ideas, why only compare with the original PPO? The authors should clearly explain the selection of baselines, and efficiency comparisons should also include this line of research.
- Furthermore, I wonder why you do not report baselines that use finer credit assignment for the DPO objective. Since you report that PPO performs better in terms of credit assignment, I am curious how it still shows superiority even when DPO is combined with finer credit assignment.
- Additionally, in practical situations, if one needs to find an optimal K for training configuration, it’s unclear whether we can say that Vine-PPO is more efficient in general, as it might require more hand-engineering. However, training the value network also requires engineering, so I wonder about the complexity comparison between these methods.

References

[1] Shao et al. DeepSeekMath: Pushing the Limits of Mathematical Reasoning in Open Language Models

[2] Ahmadian et al. Back to Basics: Revisiting REINFORCE Style Optimization for Learning from Human Feedback in LLMs

**Questions:**

- How does the method's dependency on K differ by model? I am also curious about the K ablation.
- Additionally, I think creating a graph to show the trade-off between larger K values and efficiency would be interesting.
- Very minor, but there is a missing period on line 264 (or 265).

---

> ### Author Response · Authors · 2024-11-19
> **Rebuttal**
>
> We would like to thank the reviewer for their thorough feedback. We’re thrilled that you liked our work. We now address the key questions below:
>
> **RLOO and GRPO Baselines**
>
> This is a great recommendation! As this was a common suggestion among reviewers, we posted a general comment describing RLOO and GRPO results with additional in-detail analysis. Please see *“Summary Response + RLOO and GRPO Baselines”*.  Here, we provide a short summary for convenience:
>
> RLOO and GRPO perform worse than VinePPO. On GSM8K, their respective scores of 44.5% and 44.6% fall short of PPO’s 50.1% and VinePPO’s 53.4%. A similar trend is observed on the MATH benchmark, where RLOO and GRPO achieve 17.3% and 17.8%, compared to PPO at 18.1% and VinePPO at 23.0%. Moreover, we found their training process to be less stable, requiring a higher KL coefficient during hyperparameter tuning to stabilize their training. This likely stems from their uniform credit assignment method, which contrasts sharply with the fine-grained credit assignment strategies employed by PPO and VinePPO (see “Value Prediction Analysis” in general comment).
>
> VinePPO is significantly more efficient than RLOO and GRPO, reaching their peak performance 2.7 times and 2.2 times faster on MATH, respectively. In GSM8K, which is an easier task for a 1B value network, even PPO achieves peak performance of RLOO and GRPO about 1.75x faster.
>
> Also, additional analysis included in the general comment offers deeper insights into the inner workings of RLOO and GRPO, which we highly recommend the reviewer to visit.
>
> **DPO variants with fine-grained credit assignment**
>
> There are DPO variants that aim to provide finer credit assignments, such as Self-Explore [1], as noted in our first draft (lines 132–126). Self-Explore reports an improvement from SFT (34.14%) to 37.68% on MATH using DeepSeekMath 7B [1]. In our work, PPO achieves a larger improvement, from SFT (32.8%) to 42.8% in the same setup. Given this and the engineering complexity of these methods, we decided to focus on more established baselines in the literature, such as RestEM, DPO+, PPO, and now RLOO, and GRPO.
>
> **Tuning K in VinePPO**
>
> Excellent question! Tuning VinePPO is generally simpler because it involves only one key parameter, K. In contrast, tuning the value network comes with numerous hyperparameters associated with neural network training, such as the optimizer, making it more complex. Additionally, Fig. 4 of the original paper and Fig. F.23 of the updated draft show performance and efficiency improvements as K increases, suggesting a straightforward heuristic: start with the highest K that fits within the available compute budget.
>
> **Questions**
> > **Q1.** How does the method's dependency on K differ by model? I am also curious about the K ablation.
>
> We provided an ablation study on the effect of K in Fig. 4. However, this ablation was conducted only on the 1B model due to computational constraints. Running ablation on the 7B model is prohibitively expensive. We expect to see the same pattern as increasing K always reduces the value estimation variance.
>
> > **Q2.** a graph to show the trade-off between larger K values and efficiency
>
> Thanks for the suggestion! Refer to Fig. 23 of the updated draft for this study on MATH. The results are quite interesting. VinePPO with higher K values achieves greater efficiency.  Specifically, VinePPO with K=9 is slightly more efficient than K=3, while both significantly outperform K=1. This demonstrates the strong impact of low-variance value estimates on training, shifting the trade-off towards improved efficiency with more samples.
>
> > **Q3.** missing period on line 264
>
> Thank you! We’ll fix it in the final draft.
>
> Thank you again for your valuable feedback. We hope that our response has resolved any remaining questions and concerns. Would you consider increasing your ratings given the main clarifying points outlined?
>
>
>
>
> - [1] Hwang, Hyeonbin, et al. "Self-Explore to Avoid the Pit: Improving the Reasoning Capabilities of Language Models with Fine-grained Rewards." arXiv preprint arXiv:2404.10346 (2024).

---

> ### Author Response · Authors · 2024-11-21
> **Updates and Feedback Request**
>
> Dear Reviewer,
>
> Thank you for your valuable feedback. With the discussion phase nearing its end, we’d be glad to address any further questions or suggestions to obtain a better evaluation. Importantly, we previously updated the paper with RLOO and GRPO results for the 1B model, and we’ve now added a new result for the 7B model, which completed today. Thank you again for your time and dedication to the review process.

---

> ### Comment · Reviewer_S7We · 2024-11-25
> **Response to Rebuttal**
>
> Thank you so much for the thorough response. I highly appreciate it, and I apologize for the late reply.
>
> Few Remaining Concerns:
> - According to the GRPO paper, after applying GRPO, the performance on Math exceeds 50%. Was there any difference in the settings?
> - Additionally, it seems that the benefit of GRPO diminishes for larger, well-performing models (being much more effective for the 1.1B model). Is there any specific reason for this?

---

> > ### Author Response · Authors · 2024-11-27
> > **Feedback Request**
> >
> > Dear Reviewer,
> >
> > Thank you for the insightful discussion initiated. We’re curious to know if you’ve had a chance to go over our previous response in  "Response To Remaining Concerns". We hope it has addressed your concerns. As the deadline for updating the paper is reaching soon, we’re eager to address any outstanding questions or concerns you may have, and to incorporate any additional feedback. We hope you kindly consider a renewed assessment of the paper considering our recent exchanges.

---

> > > ### Author Response · Authors · 2024-12-01
> > > **Reviewer Response Deadline Tomorrow**
> > >
> > > Dear Reviewer,
> > >
> > > Thank you again for the fruitful discussion and your technical depth in this topic. We hope our previous exchange has addressed your remaining concerns, but if there are any outstanding questions or points, we’d be eager to address them promptly especially since the deadline for reviewer response in in one day. We hope you kindly consider a stronger assessment of the paper considering our recent exchanges.
> > >
> > > Thank you for your time and thoughtful feedback!

---

> > ### Author Response · Authors · 2024-12-02
> > **Last day of reviewer response**
> >
> > Dear Reviewer,
> >
> > Thank you again for dedication to the review process. Since this is the last day reviewers can response and we addressed your mentioned concerns, we hope you kindly consider a renewed assessment of the paper considering our recent exchanges. If there're any remaining concerns preventing a stronger evaluation, we'd more than happy to address them promptly in the remaining time.

---

> ### Author Response · Authors · 2024-11-25
> **Response To Remaining Concerns**
>
> Thank you for reading our rebuttal in depth and we’re thrilled that you liked our experiments and analysis. We now answer the remaining concerns:
>
> > According to the GRPO paper, after applying GRPO, the performance on Math exceeds 50%. Was there any difference in the settings?
>
> We appreciate the reviewer's attention to details. In the GRPO paper, they use a different SFT model trained on a large (unpublished) mathematical instruction dataset containing 776K examples. This SFT model achieves 46.8% on MATH, improving to 51.7% after GRPO. In comparison, our SFT model, trained on a public MATH dataset (around 11.5K examples), scores 32.8%, which improves to 42.8% after PPO (and 46.0% after VinePPO).
>
> > Additionally, it seems that the benefit of GRPO diminishes for larger, well-performing models (being much more effective for the 1.1B model). Is there any specific reason for this?
>
> We hypothesize that larger models result in more capable value networks, which may lead to better credit assignment. As a result, PPO with value network might perform closer to or even better than methods like GRPO, which lack fine-grained credit assignment mechanisms. That said, even larger value networks are still brittle, struggling with diverse trajectories (as shown in Figure 8), questioning their true scalability.
>
> We hope that our response has addressed your remaining concerns and we are happy to engage in additional discussion if anything remains unclear.

---

### Official Review · Reviewer_wh2B · 2024-11-10

**Soundness:** 3
**Presentation:** 3
**Contribution:** 3
**Rating:** 5
**Confidence:** 3

**Summary:**

VinePPO uses Monte Carlo-based credit assignment, reducing reliance on large value networks and enhancing accuracy and efficiency. It outperforms PPO and other baselines on complex math tasks, particularly with challenging datasets. Performance improves with more Monte Carlo samples, demonstrating strong scalability potential.

**Strengths:**

VinePPO uses Monte Carlo-based credit assignment, reducing reliance on large value networks and enhancing accuracy and efficiency. It outperforms PPO and other baselines on complex math tasks, particularly with challenging datasets. Performance improves with more Monte Carlo samples, demonstrating strong scalability potential.

**Weaknesses:**

1. Lack of baselines. I suggest the author adding value-network-free methods as baselines, particularly GRPO [1] which also uses a PPO-like objective with the average reward of multiple rollouts as the baseline for the policy gradient.
2. Misuse of terminology. According to the hyperparameter setting for PPO provided in the Appendix where $\lambda = 1$ and $\gamma = 1$, PPO should produce an unbiased estimate for the value function. So it is better not to use "bias" in Line 467 and 475 but to use "inaccuracy".

**Questions:**

Questions:
1. The results show that VinePPO is quite promising for LLM reasoning, but can we extend it to the more general alignment task?
2. Is there any intuitive or theoretical explanation for why value networks fail to provide accurate estimates?

[1] Zhihong Shao, Peiyi Wang, Qihao Zhu, Runxin Xu, Junxiao Song, Mingchuan Zhang, Y. K. Li, Y. Wu, and Daya Guo. 2024. DeepSeekMath: Pushing the Limits of Mathematical Reasoning in Open Language Models. CoRR, abs/2402.03300.

---

> ### Author Response · Authors · 2024-11-19
> **Rebuttal**
>
> We thank the reviewer for their time reviewing our paper and providing useful feedback and questions which we now address.
>
> **RLOO and GRPO Baselines**
>
> As RLOO and GRPO were asked by three reviewers, we described the results with additional analysis in a general message. Please see *“Summary Response + RLOO and GRPO Baselines”* on OpenReview.
>
>
> As a short summary here: RLOO and GRPO perform worse than VinePPO. RLOO and GRPO achieve 44.5% and 44.6% on GSM8K respectively (compared to PPO’s 50.1% and VinePPO’s 53.4%) and 17.3% and 17.8% on MATH respectively (compared to PPO’s 18.1% and VinePPO’s 23.0%). Also, we found RLOO and GRPO to be less stable. For example, we found during our hyperparameter tuning that we need a higher KL coefficient to stabilize their training. When controlling for compute budget, VinePPO (and even PPO in some cases) surpasses the peak performance of RLOO and GRPO up to 2.7x faster.
>
>
> We also encourage the reviewer to refer to the additional analysis we performed in the general comment, which offers deeper insights into the inner workings of RLOO and GRPO.
>
> **Misuse of Terminology**
> > So it is better not to use "bias" in Line 467 and 475 but to use "inaccuracy".
>
> Thank you for your attention to details. We believe our use of terminology is accurate. While the policy gradient is unbiased when λ=1 (as the value estimates act only as a baseline), the value network’s estimates themselves can still be biased (see Sec. 3 of [1]) as they are approximated by a neural network (as shown in Figure 9). The term “bias” in line 467 *“PPO’s value network shows high bias.”* and in line 475 *“PPO’s value network, despite its bias,...”* specifically refers to this inherent bias in the value network’s estimation of the value, not the policy gradient. We will revise the text to clarify this distinction between bias in the policy gradient and the value estimates within the same section.
>
> **Questions:**
> > **Q1.** can we extend it to the more general alignment task?
>
> Yes. VinePPO is built on standard PPO from RLHF and does not introduce any additional assumptions to the RL formulation. So, it is broadly applicable to RL problems in language environments (as noted by reviewer S7WE), including alignment tasks where PPO is typically used.
>
> > **Q2.** Is there any intuitive or theoretical explanation for why value networks fail to provide accurate estimates?
>
> Yes. Empirically, the evidence in Section 7 (see “Error Per Reasoning Step”) suggests that the value network primarily relies on memorization, as learning a generalizable algorithm is likely less favorable given the training data and the challenging nature of the task [2]. Intuitively, the task of the value network in reasoning tasks is quite demanding: the value network must 1) implicitly understand the correct answer, 2) evaluate how the LLM’s generated solutions align with it, and 3) achieve all this in a single forward pass. This can be especially demanding given the value network is initialized from the same LLM and has similar size and capacity.
>
>
>
> We thank the reviewer again for their effort and feedback. We hope the clarifications have addressed your concerns. Given the key points outlined, would you consider increasing your ratings?
>
>
> - [1] Schulman, John, et al. "High-dimensional continuous control using generalized advantage estimation." arXiv preprint arXiv:1506.02438 (2015).
> - [2] Nagarajan, Vaishnavh et al. “Understanding the Failure Modes of Out-of-Distribution Generalization.” ArXiv abs/2010.15775 (2020): n. pag.

---

> ### Author Response · Authors · 2024-11-21
> **Updates and Feedback Request**
>
> Dear Reviewer,
>
> Thank you for your constructive feedback. As the discussion phase nears its end, we’d be happy to address any remaining questions or suggestions to obtain a better evaluation. Previously, we updated the paper with RLOO and GRPO results for the 1B model, and we now added a 7B result that became available. Thank you again for your time and dedication to the review process.

---

> ### Author Response · Authors · 2024-11-27
> **Feedback Request**
>
> Dear Reviewer,
>
> Thank you for suggesting GRPO as a baseline. We were wondering if you've had a chance to review our previous response and results regarding GRPO. Since the lack of a GRPO baseline was the primary weakness you mentioned, and our experiments demonstrate that VinePPO consistently outperforms GRPO, we wanted to ask if there are any remaining concerns that, if addressed, could help us get a stronger evaluation.

---

> ### Author Response · Authors · 2024-12-01
> **Reviewer Response Due in 1 Day**
>
> Dear Reviewer,
>
> Thank you again for suggesting GRPO as a baseline. With only one day left until the reviewer response deadline, we wonder if you’ve had a chance to review the posted results and additional analysis we performed following your suggestion regarding GRPO. Since this was the primary weakness noted, and our experiments show VinePPO consistently outperforms GRPO and is even more efficient, we wanted to know if there is any remaining concern that prevents a stronger evaluation of our work?

---

> ### Author Response · Authors · 2024-12-02
> **Final Feedback Request**
>
> Dear Reviewer,
>
> As today is the last day for responses, we kindly ask if you’ve had a chance to go over our GRPO results and whether there are any remaining concerns preventing a stronger evaluation.

---

### Author Response · Authors · 2024-11-19
**Summary Response + RLOO and GRPO Baselines**

## Summary Response

We thank the reviewers for their feedback and helpful comments.

We are heartened to hear that Reviewer S7We found VinePPO to be a “novel method in context of RL finetuning of LLMs” where “far more accurate” Monte Carlo-based estimates replace the value function achieving superior result while being “simple, easy to follow and applicable to many PPO-like conditions” — and as highlighted by Reviewer 1TWU a “very insightful” finding that the value function is an important pitfall of the PPO. We are further pleased to hear that the reviewer w2hB finds VinePPO demonstrating “strong scalability potential” while reviewer 1TWU, 3mkL, and wh2B found our experiments to be thorough “ across several tasks, model sizes, and model types” “consistently outperforming PPO and other RL-free baselines” on challenging math datasets.

We now clarify the main shared concern regarding “including GRPO and RLOO baselines” among the reviewers below and address reviewer specific questions in the individual responses.

## RLOO and GRPO Baselines

Please scroll to page 31 of the updated draft where we put all new contents for reviewers’ convenience. We will update the main content for camera-ready. The results of the RhoMath1.1B model on GSM8K and MATH are presented in appendix E of the paper (See Figure E.20, E.21). Our 7B models need more compute and powerful hardware. We are actively working on these runs and will update the paper if progress is made (*Update 11/21/2024: Added RLOO with DeepSeekMath 7B on GSM8K*).


**Result** As shown in Figure E.20, RLOO and GRPO perform worse than VinePPO. On GSM8K, RLOO and GRPO achieve 44.5% and 44.6%, respectively, compared to PPO's 50.1% and VinePPO's 53.4%. On MATH, they score 17.3% and 17.8%, while PPO and VinePPO reach 18.1% and 23.0%. These findings align with recent studies [1][2], where RLOO was found to be at best competitive with PPO (performance-wise).

**Discussion** We agree with the reviewers that RLOO and GRPO baselines are useful for practitioners and we thank the reviewers for suggesting them. Meanwhile, we think it is important to note that VinePPO is in the opposite direction of RLOO and GRPO. RLOO and GRPO remove the fine-grained credit assignment machinery of PPO, the value network, and they basically assign every token in a response the same value. On the other hand, VinePPO doubles down on fixing the fine-grained credit assignment machinery, estimating accurate values via MC samples starting from each step. (see detailed analysis below).

**Implementation and Training Details of RLOO and GRPO** Due to character limit we've put it in the first reply to this comment.

## Value Prediction Analysis of RLOO and GRPO
We follow the same protocol in Section 7 of the original draft to analyze the accuracy of value prediction for RLOO and GRPO.

**Scatter Plot + Mean Absolute Error** Figures G.24 and G.25 illustrate the distribution of value predictions across reasoning steps. RLOO and GRPO estimates show significant bias, frequently assigning high values to states with a low probability of success and low values to states with high probability. As demonstrated in Fig. G.26, although RLOO and GRPO have marginally lower MAE than PPO, their errors are still substantially higher compared to VinePPO.

**Explained Variance** Based on 1TWU suggestion, we additionally include the explained variance of value estimation in these methods. As shown in Figure G.26, VinePPO achieves higher explained variance than RLOO, GRPO, and PPO across both datasets.
Additionally, PPO’s value predictions show non-negative explained variance values close to one, indicating stable and effective training.

## Compute Efficiency Analysis of RLOO and GRPO
Following the approach in Sec 6.4 of the original paper, we plot test set accuracy against wall-clock time under the same hardware configuration to evaluate the computational efficiency of RLOO and GRPO compared to PPO and VinePPO. As shown in Figure F.22 of the updated draft, on the MATH dataset, VinePPO reaches the peak performance of RLOO and GRPO 2.7x and 2.2x faster, respectively. Notably, on GSM8K, we see the same pattern and even PPO—despite the overhead of training an additional network—surpasses RLOO and GRPO in efficiency.



- [1] Noukhovitch, Michael, et al. "Asynchronous RLHF: Faster and More Efficient Off-Policy RL for Language Models." arXiv preprint arXiv:2410.18252 (2024).
- [2] https://huggingface.co/blog/putting_rl_back_in_rlhf_with_rloo

---

> ### Author Response · Authors · 2024-11-19
> **Additional Implementation Details**
>
> ...continuation of the above comment:
>
> **Implementation and Training Details of RLOO and GRPO** For RLOO, we closely follow the implementation in the HuggingFace TRL’s library [3]. GRPO implementation is also a straightforward modification to PPO. To ensure fair comparison, we maintain equal training dynamic across runs. Specifically, we train RLOO, GRPO, PPO, and VinePPO, for 1000 iterations on MATH and 650 iterations on GSM8K (about 8 epochs for both datasets). All methods share the same rollout batch size of 512 and mini batch-size of 64. In all methods, we sample 8 responses per each question in each rollout batch. Given the 8 training epochs all methods see 64 responses per example throughout training. For all method, we initialize the policy from the SFT checkpoint. For RLOO and GRPO, we further tune the KL coefficient (search space: {1e-2, 3e-3, 1e-3, 3e-4, 1e-4}). We found that RLOO and GRPO are quite unstable and need a higher KL coefficient to stabilize their training (in our experiments 3e-3 is the smallest value they can tolerate and achieve best validation accuracy).  As a final note, the results of the RLOO and GRPO experiments look very similar. This is not a mistake. The baseline computation in RLOO and GRPO is indeed very close. Assume R1, R2, .., and Rk are task returns of K responses (Y1, Y2, .., Yk) for a prompt X. GRPO computes the baseline for policy gradient on Y1 by averaging all the returns. However, RLOO leaves the R1 out and takes the average over the remaining K-1 returns.
> - [3] https://github.com/huggingface/trl

---

### Author Response · Authors · 2024-11-22
**Updated Paper Revision**

Dear Reviewers and Area Chairs,


We summarize the updates to the paper during rebuttal, requested by the reviewers. Note that they are at pages 31 to 38 for reviewers’ convenience and the title of the sections are in blue. Specifically:

**Reviewer wh2b**:
1. Added RLOO and GRPO Baselines (Appendix E)
2. Add more details regarding terminology "bias" (Appendix I)



**Reviewer S7We**:
1. Added RLOO and GRPO Baselines (Appendix E: Figures E20, E20.1, and E21)
2. Added RLOO and GRPO Efficiency Analysis (Appendix F: Figure F.22)
3. Fixed typo (Line 264)


**Reviewer 1TWU**:
1. Added RLOO and GRPO Baselines (Appendix E: Figures E20, E20.1, and E21)
2. Added visualization of “Explained Variance” metric throughout training for all methods (Appendix G: Figure G26)
3. Added technical details of inference engines in our software stack (Appendix J)


**Reviewer 3mkL**:
1. Added More Examples of Advantages in VinePPO (Appendix H: Figures H.27, H.28, and H.29)
2. Analyzed effect of K in VinePPO’s Efficiency (Appendix F: Figure F.23)


**Additional Analysis**:
1. Added Value Prediction Analysis of RLOO and GRPO shedding light into their inner working mechanism (Appendix G: Figures G.24 and G.25)

We are eager to engage in further discussion and would greatly appreciate any additional feedback or insights, especially as none of the reviewers have not yet had the opportunity to share their thoughts at the time of posting this.

---

### Comment · Area_Chair_XVKP · 2024-11-24
**Reminder: Author-Reviewer Discussion Period Closing Soon**

This is a reminder that the author-reviewer discussion period will end on Nov 26 AoE.

Your engagement during this phase is critical for providing valuable feedback and clarifications. If you have any remaining questions or comments, please take a moment to participate before the deadline.

Thank you for your contributions to this important process.

AC

---

### Author Response · Authors · 2024-11-30
**Final Updates to the Paper - Nov 27th**

Dear Reviewers,

The results of our 7B models trained with RLOO are now finalized, and the updated paper includes RLOO results for all models and datasets. Please refer to Figure E.20.1 for the 7B results and Figure E.20 for the 1B results. These results should be particularly relevant to reviewers S7We and 1TWU, who requested the RLOO baseline.

---

### Author Response · Authors · 2024-12-04
**Final Response**

We thank all reviewers for their time and feedback.


## Summary of Discussion Period


The major concern raised by reviewers was the inclusion of RLOO and GRPO baselines. In response, **we updated the paper (added 7 pages + 11 figures), thoroughly implementing and analyzing these baselines in terms of final performance, efficiency, and value estimation accuracy.** Despite all our hyperparameter tuning (see General Response), RLOO and GRPO underperform VinePPO, even when controlling for compute. This is not a surprise and aligns well with our primary message on the importance of credit assignment. As shown in our additional analysis (Figs. G.24-G.26), RLOO and GRPO assign biased value estimates to intermediate steps, adversely affecting performance. These results are in line with recent studies on RLOO and GRPO [4,5]. Finally, besides our active participation, reviewers raised no further concerns during the discussion period.


## Recap of VinePPO


As noted by Reviewer 1TWU, many recent works [1,2,3, inter alia] attempt to simplify PPO in the context of RLHF by removing critical components, including credit assignment mechanisms, often with little to no drop in performance (reasoning tasks are an exception [3]). VinePPO is, to our knowledge, the first to address this contradiction by demonstrating that the credit assignment machinery of standard PPO (i.e., value networks) is underperforming. **VinePPO goes in the opposite direction** and attempts to fix this mechanism, demonstrating profound impacts across various axes: higher accuracy, faster convergence, better efficiency, and lower KL divergence. Here’s brief overview the credit assignment mechanisms of such methods:


| **Method**       | **Fine-grained Credit Assignment** | **Notes**  |
|-------------------|------------------------------------|------------|
| PPO (2022, [6])               | Yes                                | Trains a value network to predict the value each step during      |
| DPO (2023, [1])             | No                               | N/A       |
| RLOO & GRPO (2024, [2])      | No                               |  By design assign the same value to all steps.      |
| VinePPO (Ours)          | Yes                                |  Uses MC estimation to compute the value of each step.       |


**Primary Message of our Work**


Credit assignment, despite its importance in DeepRL, has become an overlooked aspect of RL methods for LLMs, with newer approaches often removing it entirely. Our work highlights the critical importance of this component, and we hope it encourages further research into this aspect of RL training for LLMs.


While VinePPO represents the initial attempt to principally address credit assignment in this context, it is **simple** (Reviewer S7We), **scalable** (Reviewer wh2B), and **generalizable** (Reviewer S7We).




---
1. Direct Preference Optimization: Your Language Model is Secretly a Reward Model.” by Rafailov et al, 2023
2. Back to Basics: Revisiting REINFORCE-Style Optimization for Learning from Human Feedback in LLMs, Ahmadian et al, 2024
3. SimPO: Simple Preference Optimization with a Reference-Free Reward by Meng et al. 2024
4. Asynchronous RLHF: Faster and More Efficient Off-Policy RL for Language Models by Noukhovitch et al, 2024
5. https://huggingface.co/blog/putting_rl_back_in_rlhf_with_rloo
6. Training language models to follow instructions with human feedback by Ouyang et al, 2022

---

### Meta-Review · Area_Chair_XVKP · 2024-12-21

**Metareview:**

(a) Summary of Scientific Claims and Findings

The paper introduces VinePPO, a reinforcement learning algorithm leveraging Monte Carlo-based methods to enhance credit assignment for fine-tuning large language models (LLMs) on reasoning tasks. VinePPO overcomes the limitations of Proximal Policy Optimization (PPO), particularly addressing issues with high variance and suboptimal performance of PPO’s value network in complex reasoning scenarios.

(b) Strengths of the Paper

1. The authors propose Monte Carlo-based credit assignment as a novel approach in the context of LLM fine-tuning.

2. Extensive experiments on challenging reasoning datasets (MATH, GSM8K) validate VinePPO’s superior performance and efficiency.

(c) Weaknesses of the Paper and Missing Elements

1. The initial absence of comparisons with key baselines (e.g., GRPO, RLOO) was a major concern, and while the rebuttals addressed these issues, some uncertainties about the experimental setup persist.

2. There is limited discussion on applying VinePPO to tasks beyond reasoning-intensive problems, particularly those with longer trajectories or greater computational complexity.

3. The parameter K, central to Monte Carlo sampling, requires further investigation regarding its impact on computational cost and overall efficiency.

(d) Decision and Rationale

Reviewers recognized VinePPO’s contributions to advancing reinforcement learning with human feedback (RLHF) for LLMs and its demonstrated empirical advantages. However, concerns about incomplete baseline comparisons, generalizability, and the lack of deeper theoretical insights resulted in mixed evaluations.

**Additional Comments On Reviewer Discussion:**

Reviewers appreciated the novel use of Monte Carlo-based credit assignment but asked the need for comprehensive baseline analysis (addressed in rebuttal).

Concerns about variance in MC estimates and the generalizability of VinePPO were raised, with some ablation studies partially addressing these issues.

---

### Decision · Program_Chairs · 2025-01-22

Reject